# Transformer-Based Models Are Not Yet Perfect At Learning to Emulate Structural Recursion

**Shizhuo Dylan Zhang**                                               *shizhuo2@illinois.edu*
*University of Illinois Urbana-Champaign*

**Curt Tigges**                                                            *curt@eleuther.ai*
*EleutherAI*

**Zory Zhang**                                                        *zoryz2@illinois.edu*
*University of Illinois Urbana-Champaign*

**Stella Biderman**                                                     *stella@eleuther.ai*
*EleutherAI*
*Booz Allen Hamilton*

**Maxim Raginsky**                                                     *maxim@illinois.edu*
*University of Illinois Urbana-Champaign*

**Talia Ringer**                                                         *tringer@illinois.edu*
*University of Illinois Urbana-Champaign*

**Reviewed on OpenReview:** *https://openreview.net/forum?id=Ry5CXXm1sf*

## Abstract

This paper investigates the ability of transformer-based models to learn structural recursion from examples. Recursion is a universal concept in both natural and formal languages. Structural recursion is central to the programming language and formal mathematics tasks where symbolic tools currently excel beyond neural models, such as inferring semantic relations between datatypes and emulating program behavior. We introduce a general conceptual framework that nicely connects the abstract concepts of structural recursion in the programming language domain to concrete sequence modeling problems and learned models' behavior. The framework includes a representation that captures the general *syntax* of structural recursion, coupled with two different frameworks for understanding their *semantics*—one that is more natural from a programming languages perspective and one that helps bridge that perspective with a mechanistic understanding of the underlying transformer architecture.

With our framework as a powerful conceptual tool, we identify different issues under various set-ups. The models trained to emulate recursive computations cannot fully capture the recursion yet instead fit short-cut algorithms and thus cannot solve certain edge cases that are under-represented in the training distribution. In addition, it is difficult for state-of-the-art large language models (LLMs) to mine recursive rules from in-context demonstrations. Meanwhile, these LLMs fail in interesting ways when emulating reduction (step-wise computation) of the recursive function.

## 1 Introduction

A revolution in neural methods for programming language tasks is underway. Once confined to the realm of symbolic methods, some of the most performant tools for synthesizing (Chaudhuri et al., 2021; Chen

(a) Recursion in an expression. Assume binary operators '+' and '×'. An expression can be a number or two expressions connected by '+' or two expressions connected by '×'.

Alvin enjoys fishing.
Alvin, who studies in Chicago, enjoys fishing.
Alvin, who studies in Chicago, which is a city on Lake Michigan, enjoys fishing.

(b) Recursion in natural language.This nesting of clauses is a classic example of recursiveness in a sentence, demonstrating how language can be structured in increasingly complex ways.

Figure 1: Some examples of recursive patterns.

et al., 2021; Li et al., 2022b; Chowdhery et al., 2022), repairing (Gupta et al., 2017; Saha et al., 2017; Xia et al., 2023), and even formally verifying (Agrawal et al., 2023; Yang & Deng, 2019; First & Brun, 2022; Sanchez-Stern et al., 2023; 2020; First et al., 2023) programs now rest in part or in whole upon neural foundations.

But how sturdy are these foundations? At the core of many of these tools are transformer-based large language models (Chen et al., 2021; Chowdhery et al., 2022; First et al., 2023). It is an open question to what degree these models are simply repeating program syntax, and to what degree they have some model of program *semantics*—how programs behave and what they mean. Recursive programs are one class of programs for which program semantics are especially useful and interesting. In fact, recursion is fundamental across many different disciplines beyond program semantics: It is fundamental to parsing program syntax (Figure 1a) and omnipresent in natural language (Figure 1b). It is also fundamental to logical reasoning and mathematics—the formal definition of the original Peano natural numbers, for example, is recursive in that it has a base case "one" and a recursive "successor" function, where the successor of any natural number is also a natural number.

In this paper, we investigate the degree to which transformers-based models (Vaswani et al., 2017) can learn to model a particular kind of recursion. In doing so, we join a long line of work that studies the representation capabilities of ML models through a formal language lens. The close connection between formal languages and neural network systems can be traced back to the early chapters of computer science (e.g., one key purpose of introducing regular expressions by Kleene et al. in 1951 (Kleene, 1951) was to understand the nerve nets (McCulloch & Pitts, 1943)).

Due to the omnipresence of recursion, in a broad sense, transformer-based models are already used to solve problems that rely in some way on recursion. However, these models are occasionally observed to fail in cases that demand a strong understanding of recursive structures. For example, models trained for programming tasks may misunderstand programs with nested blocks or produced imbalanced brackets. This highlights the necessity of examining the ability of machine learning (ML) models to capture recursive patterns, especially in a world where ML-assisted programming and mathematical reasoning are growing at explosive rates.

Our work focuses in particular on *structural recursion*: a restricted but powerful class of recursive functions that are defined in a structurally decreasing manner, and so must terminate. A program is an example of structural recursion if it is defined over some data structure (say, binary trees) by recursively calling itself over smaller substructures (say, left and right subtrees). Structural recursion is at the heart of important programming and formal theorem-proving tasks (which are important in software verification and mathematics) for which neural methods still lag behind symbolic methods, like inferring semantic relations between datatypes (Ringer et al., 2021; Ringer, 2021).

The difficulty of learning tasks similar to the structurally recursive ones we are interested in has manifested in prior work. For example, Zhang et al. (2022b) identify "statistical shortcuts" on simple propositional

logic, where the model fits statistical features instead of the correct reasoning steps. van der Poel et al. (2023) observe failures of neural networks on certain types of cases for regular languages, an even simpler class of formal languages than those we are interested in. Our work identifies the root of similar brittleness in the context of structurally recursive functions by further reconstructing the algorithms emulated by the learned models that lead to those errors.

Our contributions are twofold: First, **we introduce a general framework for representing and reasoning about structural recursion with sequence models.** Second, **we use our framework to conduct an empirical investigation of learned models' algorithmic behaviors on structurally recursive tasks.** Our framework and empirical results are important steps towards understanding how to better handle recursion with sequence models, paving the path towards better and more reliable performance on the wealth of tasks to which recursion is fundamental.

### Contribution 1: Conceptual Framework

Our first contribution is a comprehensive and general conceptual framework for modeling structurally recursive tasks in a way that makes it possible to analyze sequence models (like transformers trained or prompted to execute recursive computation) used to solve those tasks (Section 2). The goal of this framework is to help us connect the world of ML with the world of programming languages, so that we can clearly define what it means for sequence models to "learn to represent structural recursion."

To that end, our framework reasons about both the *syntax* (form) and the *semantics* (meaning) of structural recursion. Our framework supports syntax by way of a new sequential encoding (Section 3) based on inductive representations of recursively defined datatypes. Our framework supports semantics by way of both (a) a stepwise reduction semantics (Section 4.1) and (b) an Abstract-State Machine (ASM) semantics (Section 4.2). The reduction semantics helps us bridge the syntax with the traditional programming language semantics of that syntax; the ASM semantics makes it possible to interpret and analyze the algorithmic behavior of learned models in implementing those semantics, even when those models are not architecturally recursive.

Our framework takes into consideration the following complexities: Transformer models are not architecturally recursive—they do not call themselves. The situation can become even more complex when considering pre-trained LLMs, which are more than mere architectural frameworks—they are sophisticated systems enriched with knowledge and capabilities acquired during pre-training. In-context learning capabilities of these models add even more complexity to our goal of reasoning about recursive problem-solving capabilities. For these reasons, we design our framework to be comprehensive and adaptable, with the general principles allowing us to abstract away the complexities and focus just on the concept of recursion.

### Contribution 2: Empirical Investigation

Our second contribution is an empirical investigation of sequence model behavior on structurally recursive tasks, using our framework as a basis. We instantiate our framework to different variants of two concrete structurally recursive tasks: (a) learning the binary successor function (Section 5.1) and (b) learning various tree traversals (Section 5.2). We have chosen these tasks to be *simple* in that they demand minimal prior knowledge from the model, yet *interesting* since they capture nontrivial recursive structure.

Guided by our semantic framework, we design (Section 6) and conduct (Sections 7 and 8) experiments to better understand the behavior of small transformer models trained from scratch on variants of these two tasks. As a bonus, we also briefly investigate the ability for pre-trained language models to learn the recursive algorithm for one of these tasks from fine-tuning (Section 9). Finally, we analyze the in-context learning paradigm with LLMs (Section 10).

Our results are multifaceted. On small transformer models trained from scratch, we reconstruct the learned algorithms and find specific non-recursive shortcuts that these models take. From there, we are able to reverse engineer failure cases and understand the weaknesses of these models. Our findings suggest that transformer models trained on input-output pairs do not fully encode the recursion semantics for these tasks. Rather, they fit shortcuts pruned to edge cases, despite the possibility of constructing a transformer model to solve it up to a bounded length (Pérez et al., 2021). We use these insights to mitigate these failures.

Our fine-tuning experiments highlight potential practical issues with tokenization and pre-training data. We find the code-LMs (e.g., Code-T5) outperform general-purpose LMs with the same architecture. In the in-context learning paradigm, these weaknesses persist. Our exploration of LLM performance on these tasks (like GPT-3.5-turbo and GPT-4) shows these models' inability to mine and execute the corresponding recursive algorithms from in-context demonstrations. Our exploration also identifies common patterns of erroneous behaviors for these models. We find that LLMs struggle to find the correct rules from in-context examples and fail in interesting ways when performing in-context step-wise reduction (producing the trace of intermediate computations), like failing to terminate or executing incorrectly. Taken together, our results help bridge the theory and practice of learning recursion (Section 11) .

## 2 Framework

At the core of our work is a conceptual framework designed to systematically investigate the capabilities of models, specifically sequence models, in performing structural recursion. We first define our framework for reasoning about structural recursion (Section 2.1), then describe how we apply it (Section 2.2). Throughout, we reference Figure 2, which shows how all of this comes together.

### 2.1 Defining the Framework

Sequence models such as transformers have demonstrated remarkable capabilities in capturing patterns and dependencies in data. To delve into their ability to handle structural recursion, we separate the problem into its two core facets: syntax and semantics. *Syntax* refers purely to the form of a program (which strings of characters comprise a valid program), while *semantics* refers to its meaning. Our conceptual framework for representing structural recursion encompasses both of these and how they interact, which is crucial for a thorough analysis.

**Representing Syntax For Sequence Models: Sequential Encoding** The first challenge that our framework addresses is representing the syntax (form) of structurally recursive datatypes (like binary trees) and functions (like tree traversals) sequentially, in a form that is amenable to sequence models (Figure 2, top left). This involves mapping recursive structures to linear sequences while preserving the underlying relationships and the syntactic hierarchy. We do this in a highly general way by introducing sequential encoding adapted from the programming languages literature. The trick is to represent inputs and outputs of structurally recursive functions in terms of the language used to *inductively* construct those inputs and outputs—a trick we will describe in detail in Section 3. By doing so, bridge a well-understood representation of structural recursion in programming languages with the sequence-based paradigm of transformers, setting the stage for a detailed exploration of their capabilities.

**Representing Semantics: Bridging two Worlds** The second challenge that our framework addresses is representing the semantics (meaning) of structurally recursive functions in a way that helps us understand the learned behavior of sequence models trained on input-output pairs of syntax (Figure 2, top right). This involves bridging the common semantics of programming languages with a semantics that can elucidate model behavior. The gap between these is too wide to consider just one semantic model, so our framework considers two. Briefly, the first semantic model is a stepwise operational semantics that models the symbolic *reduction* behavior of the programming language from which we build our syntax, while the second model uses *Abstract State Machines* (ASMs) to bridge reduction with the learned behavior of the non-recursive neural models we train. We describe these both in detail in Section 4. This approach of one syntax with two semantics makes our framework useful for reconciling symbolic and learned neural behavior for structurally recursive tasks.

### 2.2 Applying the Framework

We apply our framework to create representative structurally recursive tasks and use our syntactic representation to generate data for those tasks, then use our semantics to make sense of qualitative and quantitative evaluations for those tasks.

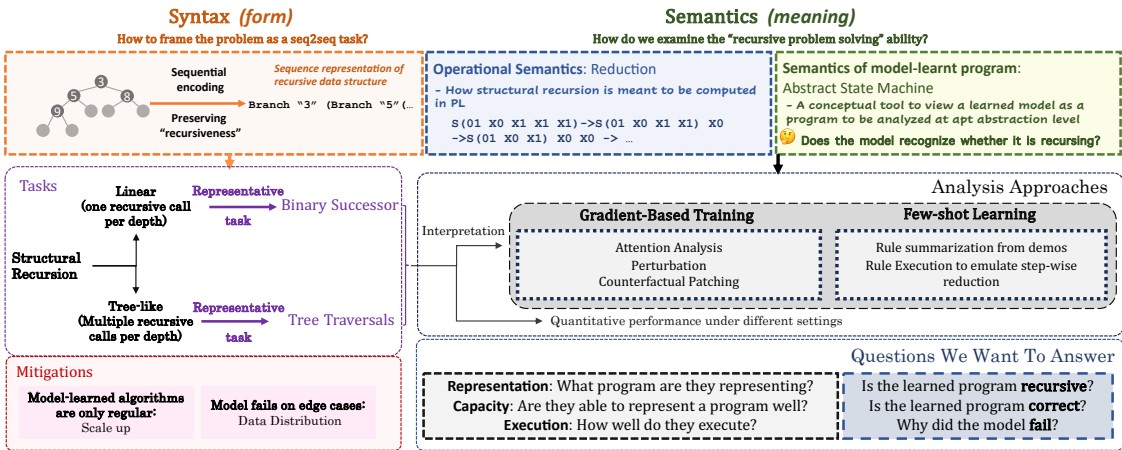

Figure 2: An overview of our framework for reasoning about sequence model performance on structurally recursive tasks, and how we apply it in this work.

**Applying the Syntax: Tasks and Data**   Our approach to contextualizing the transformers' proficiency in recursive computations is anchored in how we apply our framework's syntax to synthesize the training and testing data (Figure 2, center left). Specifically, we use the sequential encodings we have defined to craft the input/output (I/O) examples for our data. This construction of I/O examples concretizes the use of the syntax in a practical experimental context, allowing us to later use our framework's semantics to make sense of the experimental results. We instantiate this framework on two simple but interesting classes of tasks: learning a binary successor function (one recursive subcase) and learning various tree traversals (multiple recursive subcases). We describe these tasks in more detail, along with how we instantiate our syntax to these tasks, in Section 3.

**Applying the Semantics: Evaluation**   Finally, we use our framework's semantics to interpret the behavior of learned models on these tasks (Figure 2, center right). For a task that involves learning a particular structurally recursive function, our framework gives us the *desired* behavior desired behavior using stepwise operational semantics—a standard way of describing semantics for programs in a programming language. The challenge is bridging this with the *actual* learned behavior of the models trained on I/O examples. To do this, we adopt the ASM framework to reconstruct the learned "algorithm" at similar levels of abstraction to the desired recursive program. We can then analyze the learned model by looking at two key aspects: (1) the semantics of the program represented by the learned model and (2) its precision in executing that program. We devote particular attention to model behavior in edge cases, which serve as stringent tests of the robustness of the learned attempts to solve structurally recursive problems. These tell us where the models are currently falling short.

## 3   Representing Syntax: Sequential Encoding

The first component of our framework is a sequential encoding of structural recursion, which is a method of representing data in a sequence that retains its recursive structure. This approach helps us organize data points in a way that maintains their inherent recursive properties. The sequential encoding in our framework represents inputs and outputs to structurally recursive functions in terms of sequences of constructors of inductive types (Section 3.1). While these inductive types are not a new concept, our key insight is to use them to build a conceptual framework that grants us independence from character representations and underlying associations a model may have learned, and it is general enough to sequentially encode trees, language grammars, and even type checkers (Section 3.2).Furthermore, it aligns with the well-studied semantics of functions over these datatypes, which will be advantageous when we define the Section 4. All of this makes it perfect for exploring structural recursion in a sequence-to-sequence model.

```
Inductive peano =
| I : peano (* base case *)
| S : ∀ (n : peano), peano. (* inductive case *)
```

```
Fixpoint add (n m : peano) : peano :=
  match n with (* break into cases *)
  | I => S m (* if n is one, return S m *)
  | S p => S (add p m) (* otherise, recurse *)
  end.
```

(a) *Inductive type for the Peano natural numbers.*  (b) *Addition using structural recursion.*

Figure 3: Representing the Peano natural numbers using an inductive type, and writing a structurally recursive function defined over that type.

## 3.1 What: Inductive Types

> **Summary:** The representation uses the concept of induction. Think of each data point as being constructed from one or more base cases by applying inductive steps. In this way we can encode the recursive structure into a sequence representation.

Our sequential encoding employs the syntax of inductive types. An *inductive type* is a way of representing a datatype in terms of all ways of constructing elements of those types—its *constructors* (Coquand & Paulin, 1990). . There are two main kinds of constructors: some of those constructors have no recursive references to the type itself—these are called *base cases*. Others include recursive references to the original datatype— these are called *inductive cases*. These types also come equipped with *eliminators* or *induction principles* that make it possible to write structurally recursive functions and inductive proofs about those datatypes.

As an example, suppose we would like to train a model to approximate the behavior of a structurally recursive function that adds two Peano natural numbers. We start by defining the datatype representing those numbers as an inductive type in Figure 3a (the syntax here comes from a widely-used proof tool called Coq (Bertot & Castéran, 2013)).[1] This inductive type describes how to construct an instance of a Peano natural number. There are exactly two cases: (1) a single *base case* that constructs the natural number one (denoted `I`), and (2) a single *inductive case* that, given some Peano natural number `n`, constructs its successor (denoted `S n`).

We can then generate I/O examples to train a model to approximate an addition function defined over the Peano natural numbers by using a reference addition function as ground truth. In languages like Coq that have these inductive types, we can write functions (and even proofs) about these datatypes using pattern matching and terminating (structurally decreasing) structural recursion (or, equally expressively, induction). We write the reference addition function this way in Figure 3b.

Before we describe the behavior of this function, there are a few syntactic points to clarify about the language we are using: first, Coq uses parentheses only for grouping, not for function calls, so what in Python would be written `add (p, m)` is here written `add p m`. Second, the `match` statement breaks into cases based on the structure of the input: the base case `I` and the inductive case `S`. The variable `p` that comes after `S` in the inductive case is arbitrary, and matches the input, so if you pass in `S I` (two) then `p` will bind to `I` (one), and so will have that value in the recursive to `add p m`.

What that in mind, the behavior of the `add` function is as follows: We are given input numbers `n` and `m`. We break into cases based on the structure of `n`. In the base case, when `n` is `I`, we get that adding one to `m` computes the successor—that is, `add I m` computes to `S m`. In the inductive case, when `n` is the successor `S p` of some other number `p`, we get that adding the successor of `p` to `m` is the same as recursively adding `p` to `m`, then taking the successor of the result—that is, `add (S p) m` computes to `S (add p m)`.

As we will see later, we can use these reference structurally recursive functions to generate I/O examples to train our model. For example, if we call the `add` function on `S I` and `S (S (S I))`, we get `S (S (S (S (S I))))`; in other words, $2 + 4 = 6$.

---

[1]Coq was used to verify the ACM award-winning CompCert (Leroy et al., 2016) Compiler for C.

### 3.2   Why: Independence and Generality

While the Peano natural numbers are quite simple to encode sequentially without this general encoding, it turns out that we can use this same inductive representation to sequentially encode more interesting data types like binary trees and language grammars. We can also use it to draw a bridge between expected and actual model behavior and to generate I/O examples to train and test a model. What makes this choice of sequential encoding shine is its independence from specific character representations (Section 3.2.1) and its generality (Section 3.2.2).

### 3.2.1   Independence

> **Summary:** The arrangement of tokens already fully encodes the structure. How we spell these tokens does not matter.

One nice thing about this encoding and data generation process is that it is fully independent of specific character representations and any preexisting associations (for example, to numbers or trees) that a model may have learned; what we call these datatypes is irrelevant. We could just as well rename `peano` to `dinosaur`, `I` to `shampoo`, and `S` to `pencil`:

```
Inductive dinosaur :=
| shampoo : dinosaur (* base case *)
| pencil : ∀ (n : dinosaur), dinosaur. (* inductive case *)
```

and then we could `add` dinosaurs.

This independence makes this representation interesting from the perspective of investigating how models trained to learn structurally recursive functions perform without worrying about possible associations with characters that a sequence-to-sequence model may have picked up on from other training data. It will not be particularly important for our experiments on the small transformer models that we train from scratch. But it will be very important when we consider the behavior of pre-trained language models on similar tasks, as a model that has already learned associations with, say, numbers, may have misleadingly strong performance on a numeric task despite not learning the structurally recursive behavior we care about. Thanks to this independence, we can control for said associations and know when performance is truly meaningful.

### 3.2.2   Generality of Sequential Encoding

> **Summary:** This sequential encoding represents complex data types and functions while preserving all necessary information.

So far, the example of `peano` natural numbers has been quite simple. But it turns out that this simple representation is highly general. It corresponds to a broad class of datatypes well studied in programming languages, making it simple for us to define important recursive tasks. It lets us represent binary trees and even entire programming languages—all sequentially.

**Binary Trees**   First we consider encoding the type of binary trees. We start with binary trees that store character values in their nodes. We can write this type as follows:

```
Inductive tree :=
| Leaf : tree (* base case: empty leaf *)
| Branch (v : char) (l r : tree) : tree (* inductive case: branch with a character value & two subtrees *)
```

That is, a binary tree is either an empty `Leaf`, or it is a `Branch` that stores a character value `v` and two subtrees: the left subtree `l` and the right subtree `r`. We will show functions defined over this type in Section 5.

The thing to notice for the `tree` type is that, in contrast with the `peano` example, the inductive `Branch` case has *two* recursive references to the datatype—one corresponding to each subtree. In other words, binary

trees are not naturally sequential in the way that unary Peano natural numbers are. Still, by using this syntax, we obtain a sequential encoding. For example, we can represent a tree with 'a' at the root, 'c' in a node to its left, and 't' in a node to its right as follows:

```
Branch 'a' (Branch 'c' Leaf Leaf) (Branch 't' Leaf Leaf)
```

where 'a', 'c', and 't' are characters—that is, terms of type `char`. This is a sequence.

It is worth noting that, using this syntax, we could also generalize this type to represent trees with *any* kind of value, not just characters:

```
Inductive tree (A : Type) :=
| Leaf : tree A (* base case: empty leaf *)
| Branch (v : A) (l r : tree) : tree A. (* inductive case: branch with a value & two subtrees *)
```

This represents a kind of *polymorphism* (in this case, types that depend on types) (Cardelli & Wegner, 1985), insofar as the type `tree` depends on the type `A`. Here, `A` is the type of the value `v` stored in the nodes. There are infinitely many ways to instantiate `A` for such a datatype, which makes this *extremely* expressive—we can use this type to derive trees of characters (`tree char`), trees of Peano natural numbers (`tree peano`), trees of trees of characters (`tree (tree char)`), and so on.

For our purposes, it is worth noting that this expressiveness comes at a cost to the ease of data generation. After all, we could also instantiate `A` to be an uninhabitable type like `False`. In such a case, the only constructor we could call would be `Leaf`. In fact, knowing if we can ever invoke the `Branch` case of a `tree A` without knowing anything about `A` is an undecidable problem, one as general as arbitrary proof search in a higher-order logic. It is for decidability of data generation that, for the binary tree tasks in Section 5, we will stick with the first definition of `tree` and consider only character values. Still, in the remainder of this section, we will show two more examples that highlight the generality of types one can express with this sequential encoding, even when it comes at the cost of decidable data generation.

**Programming Languages** Those who are familiar with language grammars might see a similarity between these datatypes and the ways one might represent the grammar of a formal language. Indeed, it is typical to encode abstract syntax trees corresponding to parsed programs for context-free languages using these datatypes. Consider, for example, the untyped lambda calculus:

```
<expr> ::= <expr> <expr> | λ x . <expr> | x | (<expr>)
```

We can encode a datatype for abstract syntax trees for this grammar inductively:

```
Inductive expr :=
| App : expr -> expr -> expr (* e1 e2 *)
| Lam : char -> expr -> expr (* λ x . e *)
| Var : char. (* x *)
```

We could write structurally recursive functions over this datatype, like an interpreter for this small but Turing-complete language that this grammar represents syntactically. (Such an interpreter would need to be structurally decreasing over some datatype—commonly some natural number representing unbounded "fuel" the interpreter takes—and therefore terminating.) The abstract syntax trees for programs in that language would still be expressible sequentially, and could even be generated by enumerating the datatype (up to some size bound for the sake of termination). For example, the $\omega$-combinator infinitely recursive program:

```
(λ x . x x) (λ x . x x)
```

would be represented as:

```
App (Lam 'x' (App (Var 'x') (Var 'x'))) (Lam 'x' (App (Var 'x') (Var 'x')))
```

Thus, we could use this syntactic representation to encode the abstract syntax trees corresponding to arbitrary inputs and outputs to train a model to approximate an interpreter for a Turing-complete programming

language. The trouble, of course, would be that we would need to supply the necessarily-terminating interpreter with some "fuel" representing for how long it should try to interpret a potentially-nonterminating input program before giving up.

In fact, we can get even fancier—we can also use this representation to encode *relations* over datatypes. Notably, these relations can encode things like *the operational semantics of a programming language*, and so can be used to prove correctness of an interpreter like the one above. They can also encode *well-typedness relations*—giving us a sequential encoding for type derivation trees—and more.[2] This gives us an exceptionally general sequential encoding that builds bridges to two well-understood semantic models.

## 4  Representing Semantics: Bridging two Worlds

We introduce two semantic models: a stepwise reduction semantics (Section 4.1) that represents how recursive computations are evaluated step-by-step in programming languages, and an ASM semantics (Section 4.2) that allows us to analyze models trained to perform recursive tasks in the way one would analyze an algorithm. While these semantic models draw heavily on prior work, our key insight is to apply them in our conceptual framework to analyze the ideal and actual behavior of models trained to approximate the behavior of structurally recursive functions.

### 4.1  Semantic Model 1: Step-wise Reduction

> **Summary:** The desired "semantics" (meaning) of the sequential encoding is revealed through step-wise reduction, a process by which we compute the output of the recursive function step by step. Examining the models' ability to perform such reductions as subtasks allow us to evaluate whether they can represent the semantics of structural recursion.

Our first semantic model maps function inputs and outputs from their syntax (the sequential encoding from Section 3) to their corresponding semantics in the programming language from which we borrow that syntax. In the broader framework, this represents the *desired* behavior of a learned model of a structurally recursive function, though its utility is limited for understanding the *actual* learned behavior.

Our semantics of choice model the *reduction* behavior of a function applied to its inputs—a kind of evaluation behavior. We consider in particular a *small-step operational semantics* corresponding to this reduction behavior—that is, a semantics that specifies how to reduce (evaluate) a function applied to its inputs one step at a time, until what is left cannot reduce any further, at which point we have the outputs (Section 4.1.1). Importantly, this helps us decompose the task of learning the structurally recursive function into smaller pieces, making it easier to bridge theoretical and empirical behavior (Section 4.1.2).

### 4.1.1  What: $\delta\beta\iota$-reduction

At a high level, the semantics operate in terms of reduction rules (named after various Greek letters) that describe how to reduce programs that take different forms (Chlipala, 2013). There are three reduction rules are relevant to this paper:

1. $\delta$-**expansion** unfolds a constant and replaces it by its definition.

2. $\beta$-**reduction** applies a function to its arguments.

3. $\iota$-**reduction** chooses the appropriate case of pattern matching.

To turn these reduction rules into an operational semantics, we apply the reduction rules one at a time until there are no more rules to apply; this is called $\delta\beta\iota$-reduction. Note that, in our language of choice,

---

[2]We refer interested readers to the Software Foundations book series (Pierce et al., 2024) to understand how to construct datatypes for relations, along with the challenges these relations introduce to data generation.

structurally recursive functions *must* terminate, so we do not have to worry about reduction continuing forever. Note also that the question of evaluation order of these steps does not matter in this language but of course must be handled in any implementation. These semantics are well understood already, and for that we refer the reader to the relevant literature (Krivine, 2007).

For example, consider evaluating the `add` function from Figure 3b in Section 3.1. Suppose that we call `add` on the inputs `S I` and `I` to compute `2 + 1`. Using the three rules above, we can compute what we will call one "step" of computation for the sake of this paper (we take some liberty with these steps for the sake of presentation):

$$\texttt{add (S I) I} \xrightarrow{\delta\beta\iota} \texttt{S (add I I)}$$

This unfolds `add`, applies it to its two arguments `S I` and `I`, and then chooses the appropriate case. More granularly, we have:

```
                    (fun n m : peano =>
                       match n with
add (S I) I   →δ       | I => S m          →β    match (S I) with    →ι    S (add I I)
                       | S p => S (add p m)      | I => S I
                     end) (S I) I                | S p => S (add p m)
                                                 end
```

We can keep going. Composing the same three steps again, we get:

```
                    S (fun n m : peano =>
                         match n with            S (match I with
S (add I I)   →δ         | I => S m          →β     | I => S I          →ι    S (S I)
                         | S p => S (add p m)        | S p => S (add p m)
                       end) I I                     end)
```

Hence:

$$\texttt{add (S I) I} \xrightarrow{\delta\beta\iota} \texttt{S (add I I)} \xrightarrow{\delta\beta\iota} \texttt{S (S I)}$$

Note that here, we cannot reduce any further—we are done computing. This means we can say that, in *some number* of steps, `add (S I) I` evaluates to `S (S I)`. We represent this using the transitive closure of the step relation, denoted:

$$\texttt{add (S I) I} \xrightarrow{\delta\beta\iota} * \texttt{S (S I)}$$

In other words, $2 + 1 = 3$.

### 4.1.2   Why: Task Decomposition

Using a well-understood programming languages semantics of course helps us draw bridges between desired and actual behavior of the models we train. But this particular choice of semantics has an additional benefit—it allows us to break down the larger task when models struggle. The key to this is that, for every function we wish to model, we can train models for two variants of the same task:

1. one representing the transitive closure on the step relation $\xrightarrow{\delta\beta\iota} *$, and

2. one representing a single step at a time $\xrightarrow{\delta\beta\iota}$.

For a given task, the first task variant represents training a model to learn the end-to-end computation behavior of the chosen function, while the second task variant represents learning a single computation step at a time. When the model succeeds at the first task variant, the second task variant is not very important. But when the model struggles with the first task variant—as we will see for the tree traversal functions later on—this lets us break down the task into simpler subtasks.

In other words, we can think of the computation of structurally recursive functions as reduction (a single step) and composition (the transitive closure). So it is natural to ask—when a model struggles to approximate a structurally recursive function, where does it go wrong? Can it perfectly approximate a single reduction step? Can it compose two reduction steps? What about three? At what point does performance decay?

This choice of semantics gives us a way to answer these questions empirically. It also allows us to bridge the theoretical with the empirical—can we build transformer models from scratch that perform a single reduction step? Can we build transformer models from scratch that compose the right number of reduction steps? Need that number be fixed in advance? And how does this relate to what we observe empirically?

We will employ this semantic model in Section 5 to motivate our choice of task variants that we investigate and the different ways we investigate them. All of this helps us understand the *desired* behavior; the second semantic model will help us relate that to the *actual* learned behavior of the models we train.

### 4.2 Semantic Model 2: Abstract State Machines

**Summary:** Our ASM-based semantic framework is a powerful conceptual tool to view a trained model as a program. By choosing the appropriate level of abstraction, it allows us to ignore the low-level "implementation" of the program, and instead focus on the essential aspects that are useful for a mechanistic analysis of programs. This unifies the analysis of different models under different setups (e.g., prompting). Moreover, *any* mechanistic interpretability analysis involves working with an ASM at some appropriately chosen level of abstraction.

To answer the question of whether a neural model has "learned" structural recursion, an important step is to (at least approximately) recover the semantics of the model's learned algorithm. This is a nontrivial undertaking—given the complex nonlinear computations both during and after training, it is challenging to understand, interpret, and verify the exact algorithm or process the model has learned. Consequently, it is challenging to ascertain whether the model has truly learned structural recursion.

Transformer models, by architectural design, are not recursive. We thus cannot expect to easily analyze a transformer at the (relatively low) level of abstraction of internal activations and model parameters. Furthermore, there are data-driven learning paradigms that are even more challenging to reason about, such as in-context learning (by demonstration or instruction) in LLMs. Our ASM framework addresses these challenges and gives us a unified way to reason about learned model behavior.

#### 4.2.1 Concept of Abstract State Machines

The notion of an ASM (Gurevich, 1995; Börger, 2005) has emerged as a crucial tool in the world of formal methods, offering a structured yet adaptable framework for modeling of systems and algorithms. An ASM is a particular extension of the Finite State Machine (FSM) model. An FSM is written as:

$$\texttt{FSM}(in, out, \delta, \omega) = \texttt{if } defined(in) \texttt{ then } S := \delta(S, in), \ out := \omega(S, in)$$

where $S$ is the machine's state taking values in a fixed finite set of states, $\delta$ is the state transition function, and $\omega$ is the output function. An FSM has only 3 locations to be read or updated: $in$, $S$, and $out$. An ASM, in contrast, extends the notion of an FSM by admitting infinite state spaces under certain structural restrictions (see below) and by allowing the machine to read and update arbitrarily many, possibly parameterized, locations whose values can be of arbitrary type, and to have arbitrary conditions as rule guards (not only input definedness) at each step. The basic structure of an ASM is a finite set of rules of the form:

$$\texttt{if } Cond \texttt{ then } Update$$

At each time step $t$, it decides all the executable rules (i.e., those cases where the $Cond$ is true), then performs the updates under those logic blocks.

Concretely, the states of an ASM are general first-order Tarski structures (Bergman, 2015) of the form $(U, f_1, \ldots, f_L)$, where $U$ denotes the algorithm's universe that contains constants such as `true, false, undef`, and $f_i : U^{n_i} \to U$ is a finite set of functions. Some of the the functions $f_i$ may be partial Boolean

functions; they can be turned into total functions using the standard technique of setting $f_i(u_1, \ldots, u_{n_i})$ to `undef` whenever the value of $f$ at the input tuple $(u_1, \ldots, u_{n_i})$ is undefined. The purpose of these Boolean functions is to handle flow control via conditional execution according to the rule template shown above.

The generality of the definition of states is the most important aspect of ASMs. The first-order Tarski structures at their foundation represent the *most general notion* of structure that can model various structures of mathematics, abstract data types, and even key concepts of object-oriented programming like classes. It is broad enough to provide the flexibility to view a program or a system at different abstraction levels, giving us the freedom to analyze the behavior/characteristics of interest as long as one can fit the system at that abstraction level into the framework and inspect the state transitions implemented at that level.

In other words, according to the ASM framework, a suitable computation model should (1) operate at the same abstraction level as the original algorithm, (2) implement its computation through state transitions or updates, and (3) execute only a bounded number of instructions from a fixed set for state updates. ASMs satisfy these criteria and so form a perfect foundation for our second semantic framework.

### 4.2.2 Recursive ASMs

In this paper, we are concerned particularly with the behavior of a class of *recursive* programs. In this light, it makes sense to consider Recursive Abstract State Machines (RASMs)—extensions of ASMs that are specifically designed to capture recursive algorithms and processes elegantly. At the level of our framework, ASMs allow us to consider the actual behavior of transformer models as programs; RASMs help us inspect whether there is anything resembling recursion that happens inside of these programs.

The definition of a RASM is built on top of the definition of a sequential ASM (Börger & Schewe, 2020). A recursive ASM, from a high level, can be expressed in the form of "let-in" statements, each of which invokes a sub-machine that performs certain computations and directly updates the global state by returning values. This description blends in the concept of stateless recursive calls in functional programming to characterize the machine that performs recursive computation (Börger & Bolognesi, 2003).

For example, a quicksort algorithm is written as follows:

$$
\begin{aligned}
&\text{QUICKSORT}(L) = \\
&\qquad\quad \texttt{if } |L| \leq 1 \text{ then result} := L \texttt{ else} \\
&\qquad\quad \texttt{let} \\
&\qquad\qquad x = \text{QUICKSORT}(\text{tail}(L)_{<\text{head}(L)}) \\
&\qquad\qquad y = \text{QUICKSORT}(\text{tail}(L)_{\geq\text{head}(L)}) \\
&\qquad\quad \texttt{in } \text{result} := \text{concatenate}(x, \text{head}(L), y)
\end{aligned}
$$

It provides a natural sequentialization of the computation—when abstracting away the actual implementation of the QUICKSORT sub-machine, the procedure becomes "evaluate conditions, compute some values, and pass on the values to continuation."

This yields the opportunity to examine whether a sequential model (in our case, a transformer with an auto-regressive decoding mechanism) implements a recursive algorithm by inspecting the model's internals at each time step, and determining whether those internals reflect the sequence of computations needed in the recursive algorithm by an ASM.

### 4.2.3 Why ASMs?

ASMs form an appropriate basis for analyzing the behavior of learned models since they provide a computational model encompassing the original *recursive implementation* and its *approximate simulation* by a learned transformer network. ASMs are perhaps not an obvious choice for a computational model, but they are crucial to preventing loss of interpretability. In contrast, what we at first considered to be the more obvious choice methodology—situating the original recursive implementation at an appropriate level in the usual hierarchy of formal computational models and then investigating the ability of transformers to

simulate the models at that level—does not capture the relevant aspects of the problem at the right level of abstraction.

To see the loss of interpretability inherent in the above methodology relative to the ASM framework, consider modeling a function that computes the successor of a binary number. We can write a reference function for this by structural recursion in quite an elegant way, as we will see in Figure 4 in Section 5.1: We can define binary natural numbers inductively in just three cases, and define the successor by pattern matching over those three cases. Just one case requires recursive computation.

It is natural to consider constructing an iterative implementation of that recursive function using a stack, and then instantiating this iterative implementation using a Turing machine. This is doable. However, the inevitable (and unfortunate) consequence is that, since the Turing machine implementation operates at a much lower level of abstraction, the original recursive definition's elegance, parsimony, and interpretability are lost. One can appreciate this loss of interpretability by noting that, even though both the recursive and the Turing machine implementations of binary successor admit finite descriptions, it is much harder to infer program semantics from the latter than from the former. The ASM framework does not have this problem.

Another benefit of the ASM framework is that it allows us to analyze transformer models without needing a persistent state structure. Note that transformers do not implement stacks to trace recursion; instead, they are sequence models by construction. The ASM framework naturally captures this sequential structure without persistent state.

### 4.2.4 Analyzing a Learned Transformer Model with the ASM framework

In the case of analyzing transformers, the universe, $U$, can potentially include various datatype, from real matrices to token- and positional embeddings, while the functions $f_i$ might encompass operations ranging from matrix multiplications to transformer network building blocks (Phuong & Hutter, 2022; Weiss et al., 2021).

A practical example of this ASM-style update within transformers is the one-step decoding. In this process, given the string of input tokens $(u_1, \ldots, u_m)$ and the current context $(v_1, \ldots, v_k)$, a probability vector $P$ for the next token is produced. The next output token $v_{k+1}$ is sampled from $P$, thereby determining the subsequent token in the sequence. The context is then updated to incorporate this new token, resulting in $(v_1, \ldots, v_{k+1})$. This explicit representation of the one-step decoding process underlines how each decoding step in a transformer parallels an ASM update.

Two noteworthy ASM characteristics are a finite instruction set and a potentially infinite state. From this viewpoint, analyzing algorithms implemented by learned transformers entails searching for their pattern classifiers, analyzing the resulting `if-else` structures, and identifying the functions applied in each situation at every time step. A simple illustrative example of such a procedure would be "`if` {the sequence to be generated is complete} `then` generate [**EOS**] token." We omitted the underlying computation of the token probabilities inside the transformer model but only focused on the state where some boolean guards of 'sequence generation complete' are evaluated to True, and the state transition controlled by it, which is closing the generated sequence with [**EOS**].

ASM analysis can be performed in a lower level of groups of model parameters (e.g., attention heads, embedding layers, etc.) to view the model as a system and analyze how the model utilizes different parts to implement the computation. In fact, the popular concept of mechanistic interpretability, which assumes the model learned to implement some algorithm and attempts to reverse-engineer that, is exactly an exercise of ASM-style analysis of the model.

In this work, the demonstrations of I/O pairs provide a partial *extensional* description of the unknown recursive function, while the pattern classifier determining recursive calls is an *intensional* description not encoded in the training objective or data. We therefore adopt various techniques to reconstruct the ASM implemented by the model. However, in cases where accurately identifying the entire ASM implemented by the model is impossible, one could test whether the boolean guards for the invocation of recursive rules or for a neural network model, a pattern classifier, can be found as a necessary condition of the recursiveness of the approximated ASM by the model.

Finally, a few comments on ASM, program semantics, and mechanistic interpretability are in order. An ASM description of a program belongs to the domain of *operational semantics*, where one analyzes (or reverse-engineers) a program by constructing a model of a computer (or a virtual machine) on which the program runs and describing how the program "runs" on this virtual machine (Manes & Arbib, 1986). The structure of the virtual machine model reflects the chosen level of abstraction. This would involve interpreting the syntax of the program as a data-dependent sequence of instructions, which can be thought of as state transformations of an ASM. For example, replacing Peano natural numbers with dinosaurs (as in Section 3.2.1) may be thought of as replacing semantics while preserving syntax only if this replacement would also affect the abstract interpreter that "runs" the program and thus generates its operational semantics. As long as the functional form of the instruction updates is not altered, the replacement of natural numbers with dinosaurs would leave the operational semantics invariant.

Moreover, there is a natural link between operational semantics of programs and mechanistic interpretability (Olah, 2022). This is in accord with the distinction between the denotational semantics and the operational semantics of a program, where the former describes the function implemented by the program from the input-output (extensional) point of view, while the latter provides a "reverse-engineered" (intensional) view through the lens of a particular kind of a virtual machine (Manes & Arbib, 1986). In the context of neural nets, the trained Transformer provides the denotational semantics, and the goal of mechanistic interpretability analysis is to furnish an operational semantics grounded in the Transformer architecture. The ASM framework is rather flexible for this purpose and may allow for fairly complicated primitives that come up in formal analyses of programs implemented by Transformers (Phuong & Hutter, 2022; Weiss et al., 2021)

## 5 Tasks

Now that we have our framework defined, we instantiate it to model variants of two tasks: the binary successor function (Section 5.1) and a tree traversal (Section 5.2). For each task, we choose:

1. an inductive representation of a datatype (like `peano`) to use for our syntactic framework,

2. a recursive function over that datatype (like `add`) to target for learned semantics, and

3. variants of a learning task to approximate the semantics of that function.

Note that, since our interest is in whether the model can learn to emulate recursion, we train each model to approximate the function's *semantics* or behavior, rather than to explicitly represent the function's syntax or form. Nonetheless, for many tasks, it will be important also to reify these functions explicitly as syntax (for example, by synthesizing programs); we leave this to future work.

### 5.1 Single Recursive Sub-case: Binary Successor

Our first task targets the binary successor function—a function that is simple and useful, yet still structurally interesting in that it does not just amount to counting. This task is a simplification of a common benchmark used for over two decades of symbolic proof automation (Magaud & Bertot, 2000; Ringer, 2021). The target function for this task is significant in that it entirely grounds binary positive numbers in the semantics of other number formats. In fact, finding something that behaves like this function is both *necessary* and *sufficient* for aligning unary (`peano`) and binary positive natural numbers, then porting functions and proofs between the two number formats (Ringer et al., 2021). Such an alignment is non-obvious, since the structures of unary and binary natural numbers are different from one another (Ringer, 2021). This makes the binary successor function an apt target function for our first task, in that it gets us closer to discovering the semantic relations current neural tools struggle to infer.

**Datatype Syntax** We instantiate our syntactic framework with an inductive definition of a positive binary number:

```
Inductive bin_pos :=
```

```
Fixpoint s n :=
  match n with (* break into cases *)
  | O1 => XO O1 (* if n is one, return two *)
  | XO b => X1 b (* if the LSB is zero, flip it *)
  | X1 b => XO (s b) (* otherwise, recurse and shift *)
  end.
```

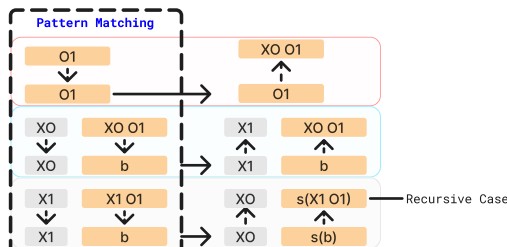

Figure 4: Code (left) and illustration (right) of pattern matching for binary successor computation.

```
| O1 : bin_pos (* base case *)
| XO : ∀ (b : bin_pos), bin_pos (* first inductive case: shift left *)
| X1 : ∀ (b : bin_pos), bin_pos. (* second inductive case: shift right and increment *)
```

That is, a positive binary number is either (1) one (the base case, denoted `O1`), (2) any another positive binary number shifted to the left (the first inductive case, denoted `XO b` for positive binary `b`), or (3) any other positive binary number shifted to the left and then incremented by one (the second inductive case, denoted `X1 b` for positive binary `b`).

One can *uniquely* construct *all* positive binary numbers by making sequences of `XO` and `X1` calls on `O1`. For example, two can be written as `XO O1`: shift `O1` to the left. Three can be written as `X1 O1`: shift `O1` to the left and then increment. Four can be written as `XO XO 1`: shift `1` to the left twice. And so on.

If we are used to thinking about binary numbers on paper, we might notice that these inductively constructed binary numbers are written backwards. We call this the "reverse" ordering. The "reverse" ordering makes it easy to define functions recursively since it is structurally decreasing—functions on sequences can be defined in terms of functions on the subsequences that follow. To recover the "natural" ordering we might write on paper, we can reverse the result and remove the `X`s. So, for example, `XO XO O1` becomes 100 (four).

**Target Function Semantics** Figure 4 contains the code for the function we target, as well as an illustration of its I/O behavior—our target semantics for this task. We target the successor function which increments any positive binary number by `1`. The function matches over the first character (the least significant bit or LSB in short) and increments. In the base case, it increments one (`O1`) to two (`XO O1`). In the first inductive case, given some binary number `b` shifted to the left (`XO b`), it increments by flipping the LSB from zero to one (`X1 b`). In the second inductive case, given some binary number `b` shifted to the left and incremented by one (`X1 b`), it increments by recursively computing the successor of `b` (`s b`), and then shifting the result to the left (`XO (s b)`). For example, the successor of eleven (1011, represented as `X1 X1 XO O1`) is twelve (1100, represented as `XO XO X1 O1`), since:

$$\texttt{s (X1 X1 XO O1)} \xrightarrow{\delta\beta\iota} \texttt{XO (s (X1 XO O1))} \xrightarrow{\delta\beta\iota} \texttt{XO XO (s (XO O1))} \xrightarrow{\delta\beta\iota} \texttt{XO XO X1 O1}$$

**Task Variants** We train our binary successor models on I/O pairs representing the computational behavior of the successor function. For inspiration and understanding, it is worth thinking through how to learn the binary successor function from examples as a human. We can learn this function from a small number of I/O examples, without invoking knowledge of numbers:

```
s O1 = XO O1                    s (XO XO O1) = X1 XO O1
s (XO O1) = X1 O1               s (X1 XO O1) = XO X1 O1
s (X1 O1) = XO XO O1            s (XO X1 O1) = X1 X1 O1
```

These examples fully represent the three cases. We compose instances using only three symbols: `XO`, `X1`, and `O1`. Heuristics and prior knowledge guide us, considering pattern matching and recursion, and preferring simple functions. We can infer a template, match cases, and fill in the template based on I/O examples.

Our goal for this task is to see how much of this semantic information a transformer model can learn *without* the priors we just described—plus how it represents the information it learns, and where the learned

algorithms fall short. It is because of this that we must generate many more than just six examples. In addition to the many examples we generate for this task, we also consider a few variants of this task. For example, we train separate models to learn from I/O examples defined using each of the "natural" and "reverse" orderings separately, to see if the ordering makes a difference. We describe more task variants as well as training details and evaluation results in Sections 6 and 7.

### 5.2 Multiple Recursive Sub-cases: Tree Traversal

For a second and more challenging task, we consider tree traversals. How transformer models approximate the behavior of tree traversals is informative for many important symbolic domains, since tree traversals are at the heart of the symbolic search procedures for programs, games, and proofs. If transformers can approximate tree traversals, this may mean better performance of neural tools for these procedures without the need for a symbolic search procedure to guide the way.

**Datatype Syntax** The representation of trees was introduced in Section 3.2.2. As a quick reminder, we consider binary trees of characters, whose syntax is represented by the following inductive datatype:

```
Inductive tree :=
| Leaf : tree (* base case: empty leaf *)
| Branch (v : char) (l r : tree) : tree. (* inductive case: branch with a character value & two subtrees *)
```

**Target Function Semantics** We target the semantics of both preorder and inorder tree traversals. An inorder tree traversal, for example, can be represented as follows:

```
Fixpoint inorder t :=
  match t with
  | Leaf => []
  | Branch val l r => (inorder l) ++ [v] ++ (inorder r)
  end.
```

Here, the base case returns the empty list `[]`. The inductive case recurses on the left and right subtrees, and composes those with the new value in the appropriate order. For example, `inorder` of the tree:

```
Branch 'a' (Branch 'c' Leaf Leaf) (Branch 't' Leaf Leaf)
```

from Section 3.2.2 evaluates to:

```
(inorder (Branch 'c') Leaf Leaf) ++ ['a'] ++ (inorder (Branch 't') Leaf Leaf)
```

which is `['c'; 'a'; 't']`.

**Task Variants** As with the previous task, we consider the problem of learning these recursive functions from I/O examples. Since the tree data structure is not sequential in its recursive structure (that is, each pass of recursion visits both the left and right subtrees), we introduce additional variants of the task that decompose these traversals into atomic computations and see if those are easier to learn. These atomic computations come from our framework's reduction semantics, decomposing recursion into one reduction step at a time. For example, to compute:

```
inorder (Branch 'a' (Branch 'c' Leaf Leaf) (Branch 't' Leaf Leaf))
```

We can run the following sequence of $\delta\beta\iota$-reductions:

```
inorder (Branch 'a' (Branch 'c' Leaf Leaf) (Branch 't' Leaf Leaf)) ─δβι→
(inorder (Branch 'c' Leaf Leaf)) ++ ['a'] ++ (inorder (Branch 't' Leaf Leaf)) ─δβι→
((inorder Leaf) ++ ['c'] ++ (inorder Leaf)) ++ ['a'] ++ ((inorder Leaf) ++ ['t'] ++ (inorder Leaf)) ─δβι→
['c'; 'a'; 't']
```

In our experiments, we look at task variants with different numbers of $\delta\beta\iota$-reductions at a time: one single step like the above, two reduction steps at a time, and so on. Our default task variant corresponds to the transitive closure of reductions. This helps us understand to what degree the difficulty stems from inferring the basic reduction computations versus from inferring how to *compose* these computations. We provide more details as well as results in Sections 6 and 8.

## 6 Experiment Set-up

We set up experiments training and fine-tuning transformer models for variants of these two tasks (Section 6.1). We also set up experiments for in-context learning using LLMs (Section 6.2) to perform both rule summarization and output prediction for variants of these tasks. This section describes the set-up for each of these experiments; results are in subsequent sections.

### 6.1 Training and Fine-Tuning Models

Our experiments training the toy models from scratch and fine-tuning pre-trained language models both used the same basic experimental set-up with a few differences. This section describes the models we trained (Section 6.1.1), the training and evaluation data we used (Section 6.1.2), and additional training details (Section 6.1.3).

### 6.1.1 Models

For each of the two tasks and their variants, we trained toy transformer models from scratch. For the binary successor task, we additionally fine-tuned pre-trained language models.

**Toy Models**  Though our framework is applicable to different architectures and modeling paradigms for sequence modeling, we focused primarily on encoder-decoder transformer models which demonstrated the strongest length generalization abilities in our set-up (see Appendix B). Our default transformer models for the binary successor task were encoder-decoder transformer models with two encoder layers, two decoder layers, a hidden dimension of 128, and 2 heads. The default model for tree traversals are also two-layer, two-head transformers, but with a hidden dimension of 256. For simplicity of the mechanistic analysis, we used sinusoidal positional encoding and greedy decoding.

**Pre-Trained Language Models**  For fine-tuning pre-trained language models, we focused primarily on encoder-decoder models, but also fine-tuned a decoder-only model. For the encoder-decoder models, we used T5-small (Raffel et al., 2020b) and T5-base, CodeT5-small (Wang et al., 2021), and ByT5-small (Xue et al., 2022). For the decoder-only model, we used GPT-2 (Radford et al., 2019).

### 6.1.2 Data

We used the same training and evaluation data for training the toy transformer models and for fine-tuning the pre-trained language models. We generated the data for these experiments using the reference functions for each task, with strategically chosen train/test splits.

**Binary Successor**  For binary successor experiments, we trained the model on all binary string pairs from 1 to $n$, where $n$ is the number of examples we wanted to use for training. For example, the maximum number we used for training is 131072, which corresponds to a 17-bit string. One can observe that the distribution of depths is non-uniform, as it aligns with the natural occurrence of different depths. In edge-case experiments in Section 7.3.2, we randomly sampled sequences from 18 to 41-bit strings to form the 'Random' test set, and collected all edge cases unseen during training to form the two edge case test groups.

**Tree Traversal**  For tree traversal experiments, by default, we split the train and test sets by tree structures, as we hoped to understand the models' abilities to deal with unseen topologies of the tree. In the experiments, we trained the models on 20,000 randomly generated trees of depths 5 and 6 and tested on 1000 held-out trees (500 each for depths 5 and 6).

### 6.1.3 Training Details

To examine the performance of our models, we computed accuracy based on the exact match of complete sequences under a greedy decoding set-up, i.e.:

$$Acc = \frac{\sum_{i \in \mathcal{D}_{test}} 1(\hat{Y}_i == Y_i)}{|\mathcal{D}_{test}|} \tag{1}$$

Length generalization was a major concern in our experimental design. In order to address the inherent limitations of transformers in generating longer sequences beyond their training exposure, we implemented a straightforward approach to eliminate the positional bias, inspired by previous research (Dehghani et al., 2019). This involved introducing a simple mechanism where we randomly adjust the positional encoding by adding random lengths of padding to the inputs. By doing so, we challenged the model to generate both the actual sequence and the corresponding padding accurately. We also applied this trick to the pre-trained language models, by having the random padding token be a new token added to the pre-trained tokenizer.[3]

### 6.2 In-Context Learning of Pre-Trained LLMs

We evaluated the ability of LLMs (GPT-3.5-Turbo and GPT-4) to execute both tasks within an in-context learning framework. Following our framework's ASM semantics, our evaluation included rule summarization as the step for us to understand the program implemented (pattern classification and function representation), as well as rule execution. Following our framework's reduction semantics, we presented the model with instructions and in-context examples in the prompt in the form of a reduction trace. For example, `(X1 X0 X1 X1 O1) = X0 (X0 X1 X1 O1) = X0 X1 X1 X1 O1` where the parentheses enclose the part to run the next step of reduction on.

**Binary Successor** As mentioned in Section 3.2.1, our framework's encoding is independent of the particular choice of characters. To rule out the effect of memorization of a certain encoding of binary numbers from pretraining, and to ensure we truly tested the model's ability to capture semantics, we also mapped the constructors to other characters (for example, $\sigma(\text{X0}) = a$, $\sigma(\text{X1}) = b$, $\sigma(\text{O1}) = c$) and compared the results with the standard choice of characters. In rule summarization experiments, we manually evaluated the model-summarized rules. In execution experiments, we checked the complete reduction chain.

**Tree Traversal** The ability to parse a tree-like structure is important for LLMs, especially considering the use cases of these models for parsing or producing structured data, as well as for planning tasks. We therefore prompted the LLM to solve the tree-traversal tasks by performing step-wise reduction. We experimented with trees from depths 3 to 6 using the same notation and presentation as in the fine-tuning experiments.

## 7 Binary Successor Experiment Results

**Summary:** The binary successor task's simplicity allowed for detailed analysis of model behavior. We conducted attention and perturbation analyses, uncovering that the model's failures on edge cases stemmed from its learned algorithm. Using data up-sampling, we mitigated these failures. Learning rate (LR) significantly impacted generalization, with different LRs leading to distinct algorithmic behaviors. The model relied more on positional information than content, especially in the natural order, and did not effectively simulate a recursive Abstract State Machine (RASM). This study reveals the model's adaptability and limitations in handling recursive tasks.

The simplicity of the binary successor task made it useful for multifaceted analytical exploration of model behavior. We first performed preliminary attention analysis (Section 7.1), followed by a perturbation analysis to recover the underlying mechanism the model learned to perform the binary successor computations

---

[3]Though the models are pre-trained on longer sequences, we observe these models to have poor generalization capabilities on this task without the random pre-padding trick.

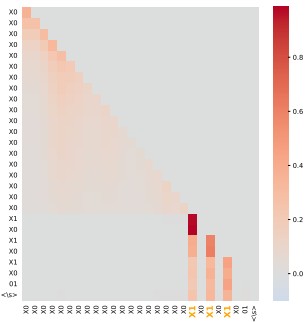
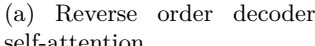

(a) Reverse order decoder self-attention.

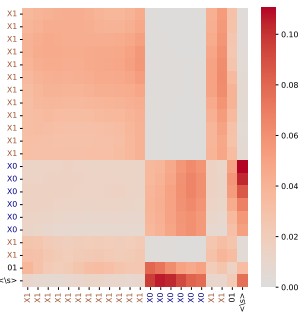
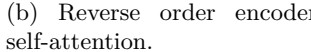

(b) Reverse order encoder self-attention.

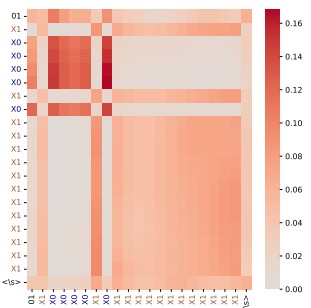

(c) Natural order encoder self-attention.

Figure 5: Attention Maps

(Section 7.2). This gave us a reconstructed algorithm that explains why the model would fail on a specific class of edge cases; knowledge that equips us to employ other techniques to overcome these edge case failures (Section 7.3). We found some additional interesting effects of the learning rate (LR) on the model's ability to accurately perform a procedure that generalizes across lengths and depths of recursion (Section 7.4). Finally, following our semantic framework, we did not find evidence that the learned model's mechanistic behavior effectively simulated a recursive ASM (Section 7.5).

Note that, in this section, we will focus solely on our analysis for this task for our toy models trained from scratch, since this was the main focus of our experimentation. We will describe our fine-tuning results for this task briefly in Section 9.

## 7.1 Attention & Embeddings

Our first step to understanding the algorithm implemented by the model was to visualize the attention maps of the model. For an encoder-decoder transformer, three types of attention can be analyzed: decoder self-attention, encoder-decoder cross-attention, and encoder self-attention. For this task, we found that cross-attention was not interesting, but decoder and encoder self-attention both were.

Our visualization of decoder self-attention revealed a noteworthy phenomenon in the final layer—something we call a *recursion head* that specializes in recursive cases. This was present in both the natural and reverse orders, though it served different purposes in each order since each order implemented a different algorithm:

- **In the natural order** (Figure 15e), the model starts attending to the last bit prior to flipping from X1 to X0, and continues to do so until the end of the sequence. Thereafter, the recursion head predominantly allocates its attention towards the token we have described.

- **In the reverse order** (Figure 5a), the recursion head directs its attention towards the X1 tokens that have been generated earlier. This attention mechanism distinguishes between recursive segments which demand rewriting, and non-recursive segments, which do not require any rewrites. The first occurrence of an X1 token encountered by the recursion head serves as a boundary that separates these distinct segments.

In summary, we use the term "recursion head" to refer to those decoder self-attention heads whose attention values were concentrated to the bit before recursion starts for the natural order, and to the last bit of recursion for the reversed order.

The encoder self-attention maps suggest that encoders also play a part in modeling semantics by helping differentiate between cases (Figure 5b and Figure 5c). In particular, the model employs its low-level attention to identify and differentiate symbols by attending to tokens of the same kind.

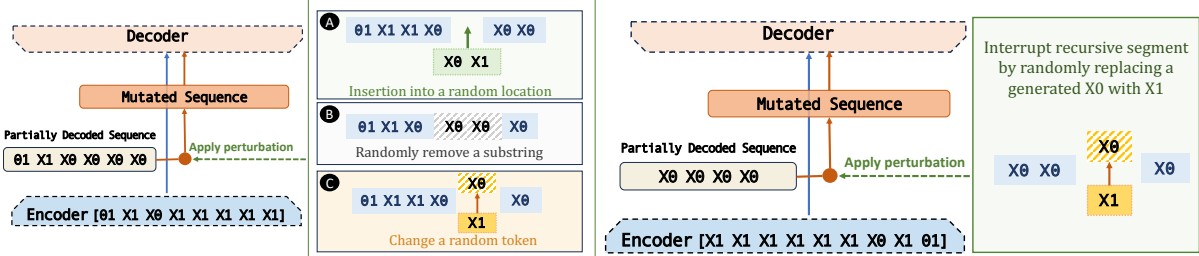

(a) String manipulation workflow for natural order.     (b) String manipulation workflow for reversed order.

Figure 6: String manipulation workflows.

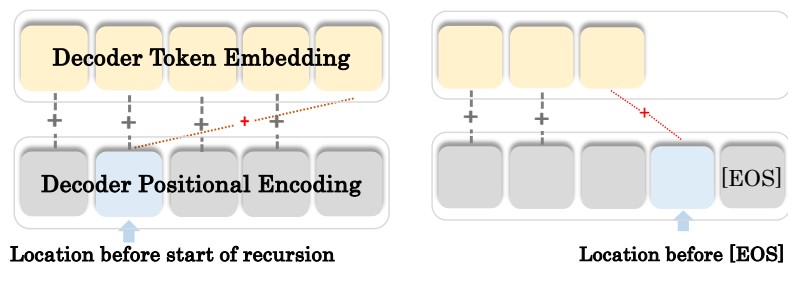

(a) Start-of-recursion Perturb.          (b) End-of-sentence Perturb.

Figure 7: Positional-encoding perturbations. 'Start-of-recursion' stands for the start of the recursion process and 'End-of-sentence' indicates the ending of a sentence.

Figures 33a, 33b, and 33c in Appendix E show T-SNE plots of the internal representation for the respective layers, with recursive segments in red and non-recursive segments in blue. The encoder (Figure 33a) distinguishes between recursive and non-recursive segments. The first decoder layer (Figure 33b) recognizes symbols. Finally, the cross-attention between the encoder and decoder (Figure 33c) injects instructions from the encoder about the start of recursion and the end of the sentence.

## 7.2   Algorithm Reconstruction by Perturbation Analysis

Attention maps are useful for forming hypotheses about model behavior, but they reveal only possible *correlations*. To understand the *mechanisms* the model employs to perform state transitions, we need to identify the relevant part of the abstract state at each time step that will determine which transition rule will be invoked at that time step.

We achieved this mechanistic understanding of abstract state by focusing on the inputs and the partially generated outputs at specific time steps while abstracting away the learned weights within the attention mechanisms of the model. In particular, we conducted perturbation analyses—mutating tokens (i.e., randomly inserting, removing, or flipping, as illustrated in Figures 6a and  6b) and swapping positional encodings (see Figures 7a and 7b) on the fly on the decoder side to see how this impacted the model's behavior. We then looked at the resulting attention maps (Figures 8a and 8b) as well as quantitative performance (Figures 9a and 9b). This experimental design was instrumental in elucidating the internal rule-based or state-transition-like mechanisms the models employed.

From this analysis, we were able to reconstruct the algorithm the model learns for both the natural and reverse orders. The pseudocode for the reconstructed algorithms can be found in Appendix D in Figures 29 and 30, respectively. Informally, the learned natural order algorithm consists of two steps:

1. It computes the total number of bits required based on the input, and identifies the position at which the last bit of the subsequence eligible for direct copying from the input sequence is located.

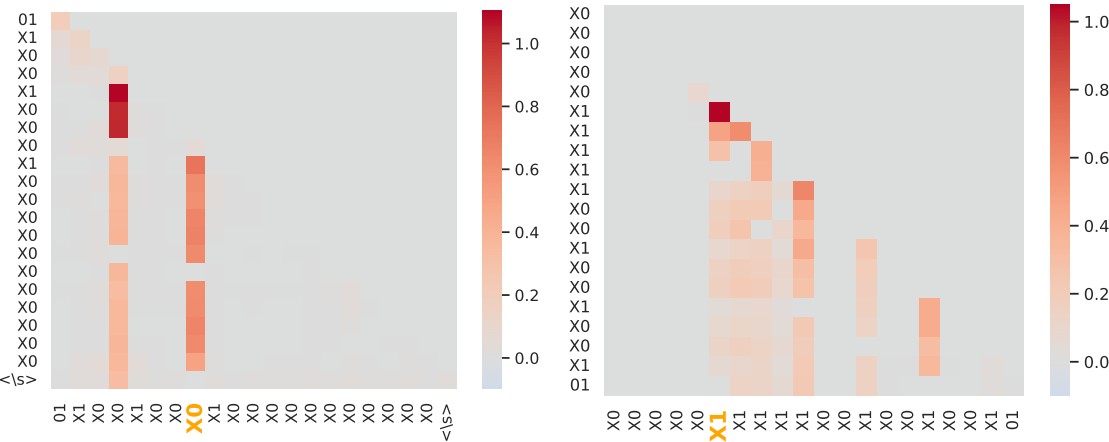

(a) Natural order start-of-recursion perturbation.  (b) Reverse order flip token perturbation.

Figure 8: Attention maps for perturbation analysis.

2. It copies this particular segment, followed by a single X1, followed by X0 tokens until it reaches the designated halting position.

Interestingly, we found that the decoder exhibits a stronger reliance on positional information rather than the content associated with each position. When we corrupted partial output using the token mutation process (Figure 6a), the model could still recover the remaining sequence (see Figure 9a). However, when we changed the positional encoding of the bit before the recursive segment to a random location (Figure 7a), the model started behaving "recursively" at the next time step by generating an X1 followed by X0s. Furthermore, if we replaced the positional encoding just before [**EOS**] with a non-terminal token, the model would immediately stop generation by producing [**EOS**]. This behavior, elucidated by perturbing token content and positional encodings, showcases the model's pronounced dependency on positional information, subtly suggesting a non-recursive, position-based logic rather than a recursive state-based one.

In the reverse order, the model behaves differently. Informally, this algorithm also has two steps:

1. Based on the input sequence, it determines the appropriate position for generating the first X1 token.

2. The decoder, while generating subsequent tokens, examines the tokens it has previously generated to determine if an X1 token has already been produced. The presence of an X1 token signals to the model to switch from generating X0 tokens to copying the remaining portion of the sequence.

We determined this by systematically replacing each X0 token within the recursive segment (excluding the last token) with an X1 token. The purpose was to observe whether the model would indeed initiate the process of copying the remaining tokens. Intriguingly, our results indicate that in approximately 93.15% of the cases, the model successfully copied the remaining tokens with complete accuracy. However, in the remaining cases, the model initially began generating X1 tokens, but exhibited confusion after a few tokens, deviating from the expected behavior. This systematic but non-recursive approach, combined with the attention map observations, subtly reinforces the idea that, while transformers may learn to perform systematic rule-based processing, the underlying algorithm they implement is decidedly non-recursive in nature.

Previous work has noted that, due to their architecture, transformer models have a structural inclination to find shortcut solutions in specific settings Liu et al. (2022). These investigations, however, have been limited to constructed models and have not explored this phenomenon in learned models. Our research addresses this void by uncovering analogous patterns in trained transformer models, specifically within the context of the complex task of structural recursion. We have also delineated these patterns more precisely by systematically reconstructing the underlying algorithms employed by these models.

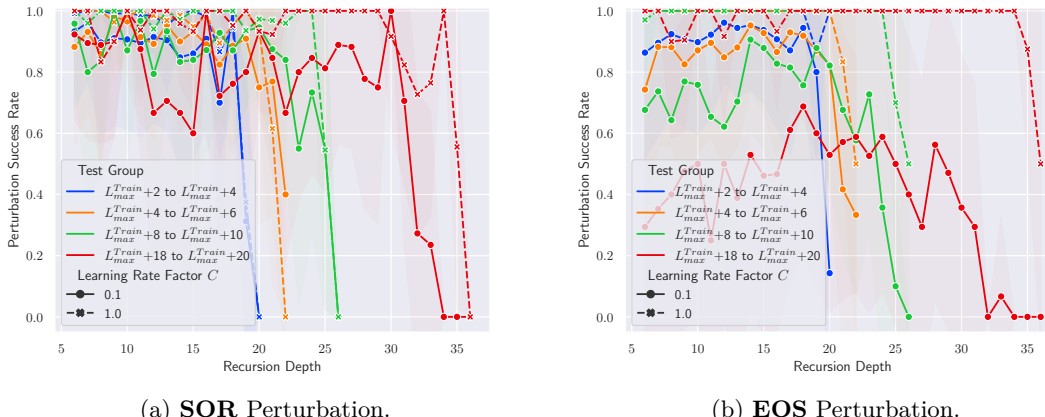

(a) **SOR** Perturbation.  (b) **EOS** Perturbation.

Figure 9: Results of positional embedding swap for natural order models. Perturbation success rate is the percentage of cases following the behavior described in Section 7.2

### 7.3 Predicting & Overcoming Edge Case Failures

We found that the models fail in interesting ways. Our reconstructed algorithms empowered us to predict specific edge cases on which the models fail, even when accuracy is high (Section 7.3.1). As these edge cases are a class of cases under-represented by the training distribution, we tested strategies to overcome them, and found that the issue can be partially overcome by data up-sampling (Section 7.3.2).

#### 7.3.1 Predicting Edge Case Failures

Though the models can achieve near-perfect accuracy elsewhere, they constantly fail in cases close to the maximum possible recursion depth for each length, especially for the natural order. Interestingly, our perturbation analysis let us *correctly predict* the cases the model fails on—and gave us an understanding of *why* that is true, too.

The specific failure cases are constructed by applying consecutive `X1` operators immediately after the `01` case. The algorithm that the model learns in the natural order falls short for these cases: it identifies the location before recursion starts and generates an `X1` followed by `X0`s, when the correct answer should be applying `X0`s immediately after `01`. In these cases, the shortcut is not applicable.

In line with our understanding of the learned algorithm, the model fails on these cases **100**% of the time for the natural order. Of all failure cases (for $C = 1$), **91**% are due to one less `X0` token generated, which is a consequence of the flaw of the model's learned algorithm. From observation, we saw that the model indeed attempts to play the same trick by finding the position right before recursion starts. However, that position is no longer within the actual sequence but rather in the "pre-padding" location. It encounters confusion between generating an `X1` or a `01` to start. It settles on `01`, but this leads it to terminate prematurely, generating a sequence that is one token too short.

#### 7.3.2 Overcoming Edge Case Failures

The edge cases that we identified in our analysis are statistically underrepresented in the training distribution, and are apparently ill-handled by trained models. A natural question that arises is whether we can train the model to learn edge cases better by putting more emphasis on these cases.

To determine this, we experimented with up-weighting and oversampling edge cases, two techniques that have been widely adopted in imbalanced classification scenarios. We used these strategies to attempt to overcome both kinds of edge cases:

1. the case where the last reduction step is `S 01` $\xrightarrow{\delta\beta\iota}$ `X0 01`, and

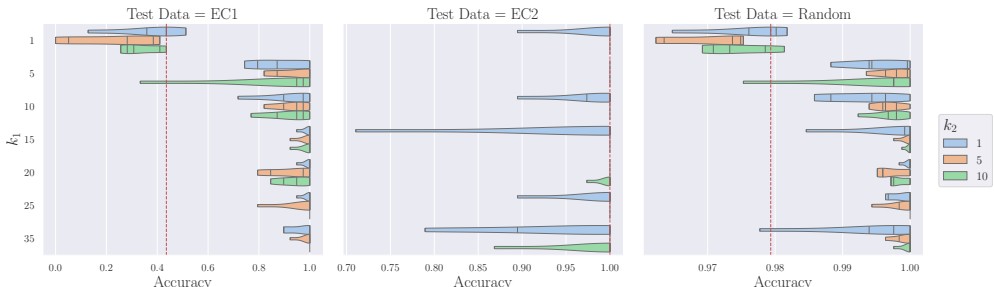

Figure 10: Reverse Order: Oversampling

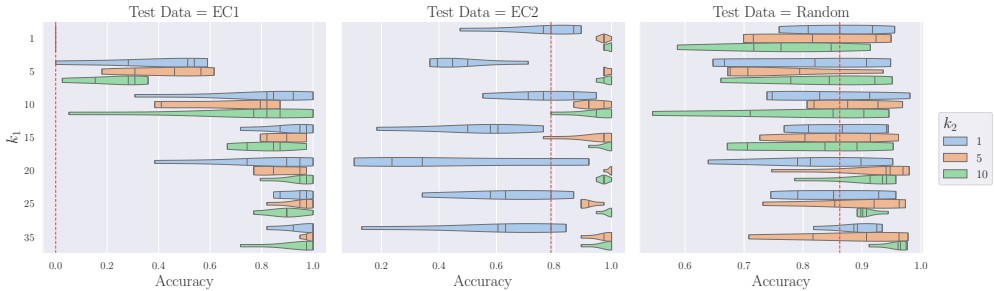

Figure 11: Natural Order: Oversampling

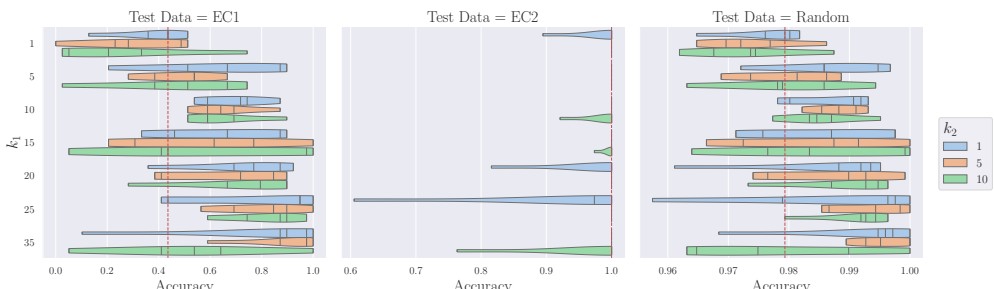

Figure 12: Reverse Order: Gradient Up-weighting

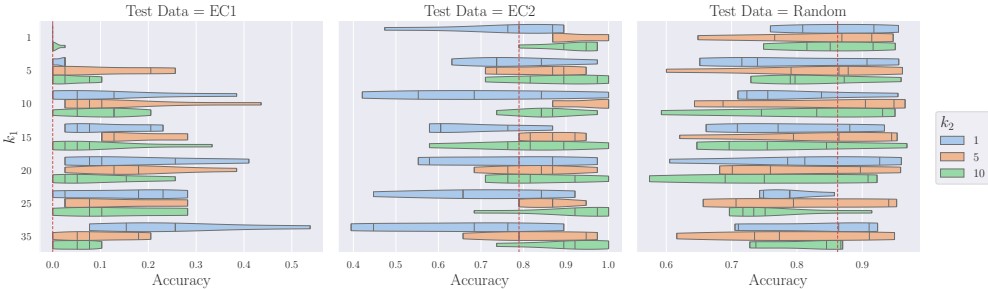

Figure 13: Natural Order: Gradient Up-weighting

2. the case where the last reduction step is $\texttt{S (X0 01)} \xrightarrow{\delta\beta\iota} \texttt{X1 01}$.

In gradient up-weighting experiments, we amplified the gradients with respect to each of the two kinds of edge case (if they occurred) by a factor we denote $k_i$ for edge case $i$ (i.e., $k_1$ for edge case 1, and $k_2$ for edge case 2). In oversampling experiments, we duplicated the edge cases of kind $i$ $k_i$ times in the training data.

We found that the oversampling strategy *did* significantly help the model improve on capturing the edge case patterns with *minimal influence* on the non-edge case data points for both orders (Figures 10 and 11). In

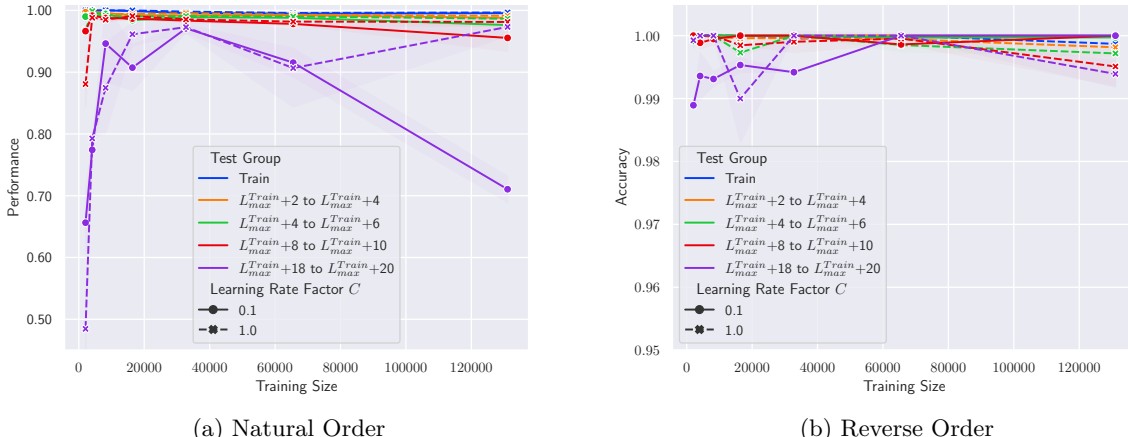

| (a) Natural Order | (b) Reverse Order |

Figure 14: Performance versus number of training examples. The performance is the average of samples with all possible recursion depths with in that length range. The error bar indicates the standard deviation across total of 3 runs with different random seeds.

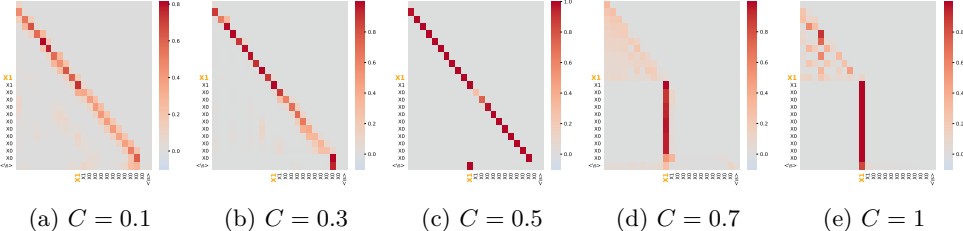

(a) $C = 0.1$    (b) $C = 0.3$    (c) $C = 0.5$    (d) $C = 0.7$    (e) $C = 1$

Figure 15: Self-attention of the last decoder layer under different LR factor $C$'s on natural order.

contrast, while gradient up-weighting also helped in the edge cases, it had a *negative impact* on the models' performance in non-edge cases (Figures 12 and 13).

It is important to note that one cannot identify these edge cases beforehand because doing so requires an understanding of the algorithm that the model has learned. This unveils the brittleness of mining algorithmic rules from demonstrations, even with a representation that inherently encodes the structure.

### 7.4 Generalization

One interesting thing that we observed in our experiments is that LRs affect model precision for the binary successor task. Thus, LRs also affect model generalization ability across lengths and to edge cases. In fact, it appears that changing the LR can even impact the *intensional behavior* (that is, the particular algorithm) that the model learns!

To determine this, we varied the LR following the original transformer LR scheduling scheme (Vaswani et al., 2017): $\alpha = C * d^{\frac{1}{2}} * \min\{s^{-\frac{1}{2}}, s * S_w^{-\frac{3}{2}}\}$ where $d$ is the embedding size of the model, $s$ is the current number of update steps, $S_w$ is the predefined warm-up step number, and $C$ is the constant controlling the magnitude. The overall generalization performance across different LRs is in Figures 14a and 14b.

To our surprise, we observed a significant difference in attention patterns when trained with different values of $C$ on the natural order task, which could suggest behavioral differences in the model that might result in different generalization abilities. As the LR grew, the model began to specialize one head into a recursion head gradually, as shown in Figure 15. In fact, smaller LR models learn the weaker notion of executing the algorithm in Section 7.2 for longer sequences (see Figures 9a and 9b).

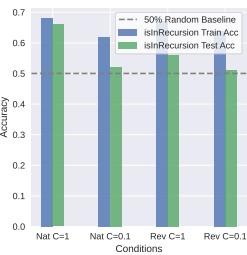

(a) Probing average-pooled embedding to determine whether the transformation needs recursion.

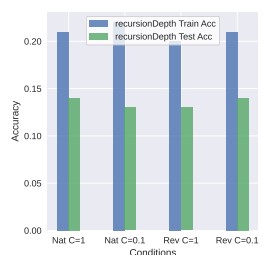

(b) Probing average-pooled embedding to determine the maximum depth of recursion.

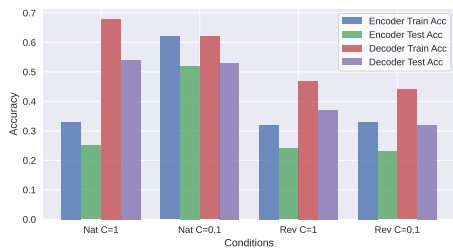

(c) Probing each token embedding to determine the current depth of recursion.

Figure 16: RASM-Probing results.

We also looked at varying LR while constraining the training dataset to contain only shallow recursive depths for the binary successor task. In this case, we found that model trained on a small LR sees a steeper drop in test performance while still attaining near-perfect training accuracy. However, for the reverse order, such a depth constraint does not severely affect the model's performance on either LR (see Figure 25 in Appendix D.1).

### 7.5 Probing: How Effectively Did the Transformer Simulate an RASM?

In alignment with the abstract state machine (ASM) framework discussed before, our empirical evaluation aimed to determine if trained transformers emulate a recursive ASM (RASM). One of the primary objectives of our study was to investigate whether the model implicitly implemented a recursive algorithm by pattern matching on the most significant bit, thereby simulating an RASM. To accomplish this, we conducted an initial analysis to examine the embeddings of the encoder and decoder for pattern-matching and recursion-depth-related information. Our ideal outcome was to identify evidence supporting the implementation of a recursive Abstract State Machine (RASM) described in Section 4.2.2 as employing a pattern-matching scheme. Alternatively, we expected that a sequentialized algorithm for the recursive task might exhibit indications of tracking the depth of recursion.

To delve deeper into this matter, we designed a straightforward experiment. We sought to train a simple linear classifier capable of distinguishing between cases such as `X0 bb` and `X1 bb` based on the model's internal representations. We computed the average encoder output across the sequence and trained a linear classifier on the fixed encoder representations for binary (whether recursive computation is required) and recursion-depth (the number of reductions needed to compute the successor) classification. The outcomes of these experiments are presented in Figure 16a. Regrettably, our attempts to train a classifier to detect the distributional distinctions in the embeddings between the two cases proved unsuccessful.

Next, we aimed to investigate the extent to which the embeddings of individual tokens contained information regarding the recursion depth. We found some evidence of this, but only in the natural order. To achieve this, we trained a classifier to recognize the recursion depth of each token using both the encoder embeddings and the output embedding of the final decoder layer. The results, presented in Figures 16b and 16c, demonstrate that the encoder embeddings do not contain information on the current depth of recursion, while the decoder embeddings *do* exhibit a certain degree of such information in the natural order. However, we did not observe this phenomenon when considering the embeddings in reverse order.

All of this evidence suggests that the transformer models we have trained do not strictly implement the kinds of state transitions one might associate with an RASM. The encoder's absence of positional information and

the decoder's limited positional information in the natural direction further underscore this deviation from the expected recursive behavior.

## 8 Tree Traversal Experiment Results

**Summary:** The transformer models trained from scratch fail to learn a recursive solution to tree traversal. They learn shortcuts from data for pre-order traversals, and learn depth-specific approaches for step-wise reductions, making generalization to unseen depths impossible without additional layers and heads.

The tree traversal task is inherently more complex and challenging than the binary successor task, since performing each reduction step in a tree traversal requires more sophisticated logical reasoning. To recover the algorithmic behavior of the model, we again performed an ASM-guided mechanistic analysis (Section 8.1). Though different traversals are of similar levels of difficulty, we found that the model still tends to pick up on shortcuts if possible (Section 8.2). The shortcuts we reconstructed (Section 8.3) are for fixed depths (Section 8.4), and do not strictly model recursion.

Taken together, our findings reveal that the models focus their attention on different components of an ASM-style update (i.e., constituting logical guards and implementing the transitions), and that they exhibit distinct behaviors depending on the kind of traversal and the number of reduction steps (for step-wise reduction). The model picks upspecific solutions for solving the cases for a given depth, which aligns with the formal language intuition of parsing using regular grammar (without counting).

### 8.1 Algorithm Reconstruction by Counterfactual Patching

Analyzing models trained for the tree traversal task is more complicated than analyzing models trained for the binary successor task. We can see this by looking at both the ASM and reduction semantics in our framework: First, the ASM corresponding to each model trained for this task, in order to produce one symbol, needs to handle composite logic formulas and invoke multiple mechanisms (e.g., pointing to correct locations and extracting values). Second, each $\delta\beta\iota$-reduction step in our reduction semantics for this task implicitly captures not just one substitution, but rather two concurrent substitutions—one in each of the left and right subtrees.

It is because of this complexity that, in order to analyze the intensional behavior of the learned model, we designed additional subtasks that broke down even a single reduction step further, in ways that align more naturally with learned transformer model behavior. Our process for analysis was the following:

1. decomposing the tree traversals and their reductions into these subtasks,

2. conducting "activation patching" in order to determine what parts of the network were most important to accomplishing each subtask, and

3. examining the attention pattern behavior for the most significant attention heads, as well as the patterns of earlier heads that likely contributed to each given head's behavior.

There are seven total subtasks that we tailored for this analysis, which are designed capture the key 'transition' points in performing tree traversal. Not all applied to every case (e.g., some were relevant for only one of the preorder or inorder traversal), but overall these subtasks include:

- **Copy initiation:** the point at which the model outputs the first token of a copy of a sub-sequence of the input sequence.

- **Midpoint of left tree copy (for in-order traversal and reductions with depth $\geq 2$):** the point at which the model outputs the root of the left sub-tree in an in-order traversal. We want to observe how the model fetches the right sub-tree

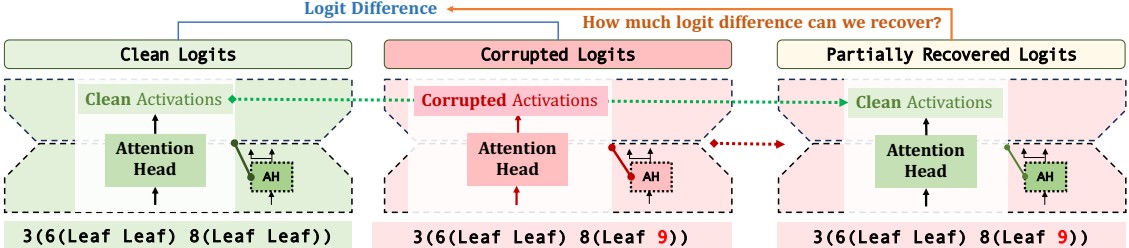

Figure 17: Illustration of counterfactual patching process.

- **End of left tree copy (for inorder traversal):** the completion of the in-order sequence of the left tree.

- **Insertion of root node:** the point at which the model outputs the root node from the top level of the tree.

- **Insertion of REDUCE (for reductions):** in partial traversals, the point at which the model outputs the REDUCE token used to demarcate an untraversed sequence.

- **Resumption (for reductions):** the point at which the model continues to copy the sequence after inserting REDUCE.

- **EOS:** completion of the sequence.

To identify the most significant attention heads to the models' final output, we adopted an intuitive yet useful approach inspired by Meng et al. (2023) and Wang et al. (2023a) called counterfactual patching (Figure 17). From a high level, this method determines the importance of an attention head by replacing an activation when producing answer **A** with one that is obtained when the model is used to produce a different answer **B** for a different input, then observing how much the answer is drifted from A to B.

To perform this process, we first perform a **clean run**, where we directly feed the input sequence to the model as normal and collect the activations and logits $l$. Then we perform a **corrupted run** by flipping a token which is tested to indeed influence the model outputs to obtain the corrupted logits $\tilde{l}$. We then run the model on the corrupted embeddings while replacing the hidden state of some token on a particular head with the one from the clean run. The effect will propagate to the subsequent layers. To measure the importance of this head, we compute to what degree the logit difference between $l$ and $\tilde{l}$ can be recovered by bringing back the clean activation from that particular attention head. The extent to which we can recover the logit difference reflects the importance of this attention head.

We used this technique to determine which attention heads were most important to the final output of the transformer. This did not reveal the full circuit used to perform the task, but it did provide suggestive, causal evidence of what the learned model is doing. After identifying these key components of the model, we focused our investigation on the roles and composition behavior of these attention heads, leaving a comprehensive analysis of the entire circuit for future research.

## 8.2 Identifying Learned Shortcuts

As shown in Figure 18, we found that models can perform full preorder traversals on unseen trees, but fail completely for inorder traversals. Even more interestingly, the model can learn the full preorder traversal better than it can learn the step-wise reductions—because it learns shortcuts.

Examining the attention behavior of the model, we observed that the model primarily focuses on numerical values and disregards brackets and EMPTY tokens in preorder traversals, as observed through its cross-attention (see Figure 31b in Appendix D.2). The learned transformer attends only to node values while ignoring

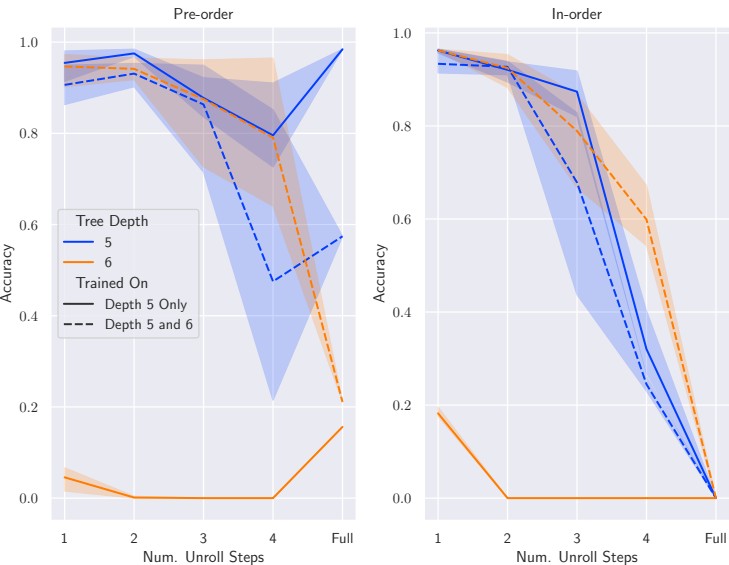

Figure 18: Accuracy of tree traversal.

parentheses and leaf nodes in first layer cross-attention, and additionally acquires information on the next token to be copied (i.e. the next node value token), shown by cross-attention in the second layer.

For inorder traversals, we hypothesize that there is no clear shortcut for sequence models to perform inorder traversals without stacks or more sophisticated architecture (with more layers or attention heads). Unlike preorder traversals, which can be done linearly, inorder traversals demand "understanding" and capturing recursive relationships between nodes.

### 8.3 Reconstructed Algorithms

The learned transformer model employs a set of logical rules and functions facilitated by the attention mechanism for tasks such as copying tree node values or producing special tokens. It uses various types of attention to keep track of important token locations for either logical control flow or invocation of the mechanisms to produce the next token in the output sequence.

In the task of inorder twice reduction, for instance, the model uses its attention to identify the appropriate reduction level and pinpoint the node whose values will be copied to the current output slot (see Figure 32b in Appendix D.2). The model combines multiple logical conditions to discern the boundary separating the remaining sub-trees and the layers set for unrolling. To generate the `"UNROLL["` statement in preorder reduction, the model's cross-attention is focused on three key positions: 1) the parenthesis immediately preceding the node where the `"UNROLL["` token will be inserted, 2) the subsequent opening parenthesis, and 3) the final closing parenthesis of the subsequence that will be copied into the `"UNROLL[ ]"` statement.

### 8.4 Fixed-Depth Nature of Model-Learned Solutions

In the context of formal language theory, parsing a tree involving a fixed number of reduction steps using regular grammar is known to be feasible. This process can be executed seamlessly by predefining the handling of each possible pattern, thus simplifying tree parsing. Similarly, our experimental results show that transformers have demonstrated the capability to learn to parse reductions of a specific number of steps. They follow a similar route by learning to identify patterns inherent to that particular depth. As illustrated in Figure 19, the model's learned solution peels off the fixed number of parentheses to locate the root of the tree to run reduction on.

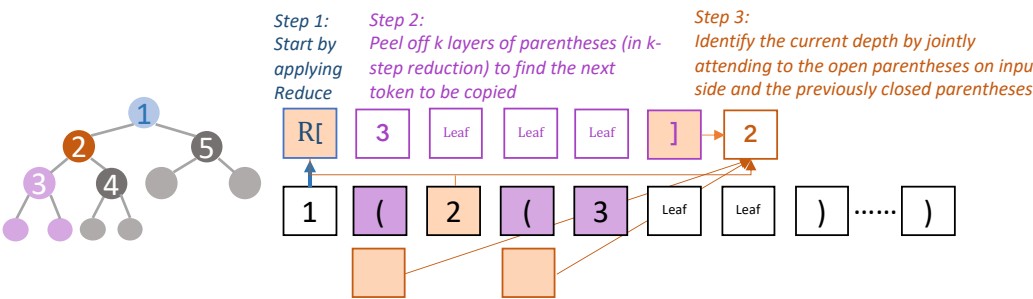

Figure 19: The illustration of how the model performs reduction with its attention in inorder traversal. The example is a two-step reduction for the given tree.

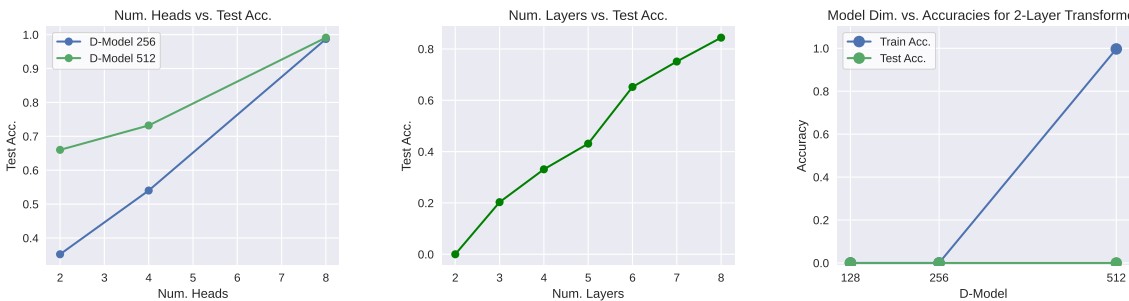

Figure 20: Results in up-scaling trees. In the number of heads versus test accuracy experiment, we experimented with 4-layer transformers. To observe the trend with an increasing number of layers, we fixed the hidden dimension to be 256.

Yet a vital limitation of regular grammar emerges when it comes to parsing a tree of arbitrary depths. To accomplish such a task, the need for a counter becomes essential. Pressing further into the complexities of depth increases, the model requires a larger number of heads. Each head gathers a specialization to manage parentheses of different depths. However, given that there is no clear constant-depth workaround for this task, and considering the absence of a counter construct in the learned transformer, a performance decrease with increasing depth is anticipated.

From the RASM perspective, a noteworthy feature of the learned algorithm becomes clear, too. In particular, the learned model does not implement a modularized algorithm that could be viewed as two independent agents of the RASM.

Our experiments to change the model architecture by increasing the number of layers, hidden dimensions, and attention heads verified the reasoning in the previous subsection (see Figure 20). We did see an increase in test accuracy with the increase in model depth and embedding size, which was natural to anticipate. We also found that, with the depth and embedding dimension held constant, simply increasing the number of attention heads allows the model to parse more complex cases. As the maximum depth of trees increases, models need to handle more different depths and thus need more heads to be specialized to each. Indeed, as we increase the number of heads given the same embedding size, the performance increases.

These findings echo our earlier discussion on the fixed-depth nature of the algorithm transformers learn. We conjecture that the key to a generalizable algorithm could be the specialization of some parameters to implement a counting mechanism that tracks the depth of parenthesis, which may be hard to learn from I/O examples of tree-traversal functions or their reduction semantics alone using the existing transformer architecture.

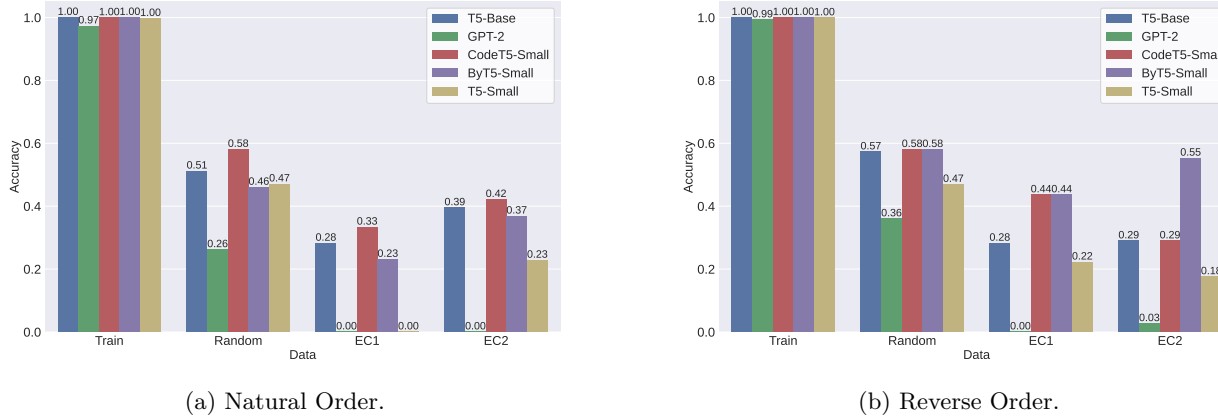

(a) Natural Order.

(b) Reverse Order.

Figure 21: Performances of pre-trained models on natural and reverse order traversals.

# 9 Can a Pre-Trained LM be Fine-Tuned to Perform Structural Recursion?

**Summary:** Pre-trained LMs finetuned on the binary successor tasks also struggle to generalize and the performance varies.

One step beyond working with toy models, we briefly investigated fine-tuning pre-trained models to learn structural recursion. Our goals were to test whether the associations established by pre-training help with picking up the notion of recursion as found in Zhang et al. (2022c) and to identify the potential practical issues with pre-trained transformer models in performing symbolic tasks. We tested the models' performance on the binary successor task; results are in Figure 21.

Similar to our experiment findings on length extrapolation with toy models in Appendix B, we found that the decoder-only pre-trained GPT-2 model tends to generalize worse than T5 models, though achieving near-perfect accuracies on the train set. By comparing various T5 models, we found ByT5-small outperforms T5-small models since it allows the model to better by-pass the meanings of specific tokens (e.g., X, 0, 1) and pick up the semantics of transformation from pairs of inputs and outputs. Pre-training over a code corpus equips CodeT5 with better-structured reasoning capabilities and the inherently recursive and tree-like structures of programs allow it to better solve problems recursively. Therefore, CodeT5-small achieves better overall performance. Another interesting observation revealed from the accuracy results is that the reverse-order representation is easier to learn by both pre-trained and toy models than natural order.

By comparing performances of T5-small and T5-base models, we found evidence for pre-trained models to learn this simple task better—even though a 2-layer toy transformer is enough to achieve high accuracy, as we showed in Section 7.

# 10 In-Context Learning Results

**Summary:** This section evaluates GPT-3.5-Turbo and GPT-4 in in-context learning for structural recursion tasks. They struggle with maintaining algorithmic consistency despite detailed prompts, often making errors in recursive rule execution and termination. The experiments reveal that while in-context examples offer some guidance, they are insufficient for the models to learn correct recursive rules. Notably, the models perform better without in-context reductions, relying more effectively on pre-trained knowledge. This underscores the need for improved mechanisms to enhance their consistency and accuracy in handling complex recursive tasks.

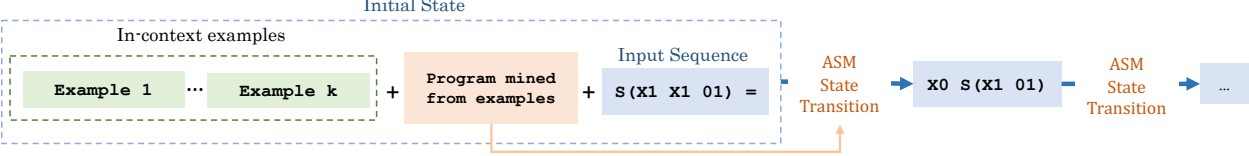

Figure 22: Example Analysis of In-Context Learning

| Order | Tokenization | Examples | CoT | #Examples | Model | Hit@1 | Hit@3 | Hit@5 |
|---|---|---|---|---|---|---|---|---|
| Natural | Original | inorder | Full trace | 10 | GPT-3.5-Turbo | 0.10 | 0.25 | 0.25 |
| Natural | Original | inorder | Full trace | 10 | GPT-4 | 0.30 | 0.40 | 0.60 |
| Natural | Original | inorder | No CoT | 10 | GPT-3.5-Turbo | 0.10 | 0.20 | 0.30 |
| Natural | Original | inorder | No CoT | 10 | GPT-4 | 0.20 | 0.30 | 0.30 |
| Reverse | Original | inorder | Full trace | 10 | GPT-3.5-Turbo | 0.10 | 0.15 | 0.20 |
| Reverse | Original | inorder | Full trace | 10 | GPT-4 | 0.45 | 0.55 | 0.60 |
| Reverse | Original | inorder | No CoT | 10 | GPT-3.5-Turbo | 0.05 | 0.15 | 0.25 |
| Reverse | Original | inorder | No CoT | 10 | GPT-4 | 0.20 | 0.20 | 0.20 |
| Natural | Original | Random | Full trace | 10 | GPT-3.5-Turbo | 0.25 | 0.35 | 0.45 |
| Natural | Original | Random | Full trace | 10 | GPT-4 | 0.10 | 0.50 | 0.50 |
| Reverse | Original | Random | Full trace | 10 | GPT-3.5-Turbo | 0.25 | 0.45 | 0.50 |
| Reverse | Original | Random | Full trace | 10 | GPT-4 | 0.00 | 0.30 | 0.40 |
| Natural | Permuted | inorder | Full trace | 10 | GPT-3.5-Turbo | 0.10 | 0.20 | 0.20 |
| Natural | Permuted | inorder | Full trace | 10 | GPT-4 | 0.20 | 0.30 | 0.40 |
| Reverse | Permuted | inorder | Full trace | 10 | GPT-3.5-Turbo | 0.10 | 0.30 | 0.30 |
| Reverse | Permuted | inorder | Full trace | 10 | GPT-4 | 0.30 | 0.40 | 0.50 |

Table 1: Learning to perform binary successor task with in-context examples. The text "inorder" in the "Examples" column means providing the model with examples starting from base case (i.e. the successor of $1_b$ to $10_b$) in order. "Random" means randomly sampling 10 number-successor pairs.

For in-context learning of LLMs, one can choose different levels of abstraction depending on the purpose of the study. In our case, we analyze the model from the text level, where we consider the model performing step-wise computation of the program in-context, and the current state is the context at that time step. We treat the programs the LLMs summarized from in-context examples to be the "learned" program of the model that it will subsequently execute, and we measure its ability to execute structural by producing the trace of reduction conditioned on examples and on the set of transition rules as instructions.

At this point, the flexibility of the ASM framework offers us the freedom to lift the level of abstraction where we assume the model is executing the program (including the pattern classification and transition) as explicated in the text using its internal mechanism (see Figure 22). This allows us to unify the analysis framework of learned models via gradient updates and in-context learning.

Our rule-finding experiments show that the LLM under test (GPT-x) cannot summarize the correct recursive rules from the various combinations of in-context examples effectively, with an observed accuracy of only 18.75%. The pattern classifiers they learn from in-context examples are superficial, even with hints restricting the number of rules and how the rules should look, and non-exhaustive (the rules do not cover all cases in the in-context examples).

From Table 1, we notice the model lacks the capability to accurately compute binary successors under various set-ups. Nevertheless, teaching the model to perform step-wise reduction to reach the final result *does* help the model solve more cases correctly by performing a simpler operation each time and allowing the model to store intermediate computations in the context.

In addition, when presented with the state transition rules and step-wise reduction demonstrations in a chain-of-thought style in the prompt, the LLM still struggled to execute the recursive rules correctly to produce the successor of the test input. Some typical errors included accidental missing or inserting tokens,

| Depth | Model | No Reduction | | | Single Step Reduction | | | Full Reduction Trace | | |
|---|---|---|---|---|---|---|---|---|---|---|
| | | Hit@1 | Hit@3 | Hit@5 | Hit@1 | Hit@3 | Hit@5 | Hit@1 | Hit@3 | Hit@5 |
| 3 | GPT-3.5-Turbo | 1.00 | 1.00 | 1.00 | 0.30 | 0.65 | 0.70 | 0.10 | 0.45 | 0.55 |
| 3 | GPT-4 | 1.00 | 1.00 | 1.00 | 0.95 | 0.95 | 0.95 | 1.00 | 1.00 | 1.00 |
| 4 | GPT-3.5-Turbo | 1.00 | 1.00 | 1.00 | 0.35 | 0.65 | 0.75 | 0.25 | 0.45 | 0.60 |
| 4 | GPT-4 | 1.00 | 1.00 | 1.00 | 0.80 | 0.85 | 0.95 | 1.00 | 1.00 | 1.00 |
| 5 | GPT-3.5-Turbo | 0.70 | 1.00 | 1.00 | 0.35 | 0.65 | 0.70 | 0.05 | 0.10 | 0.15 |
| 5 | GPT-4 | 1.00 | 1.00 | 1.00 | 0.95 | 1.00 | 1.00 | 0.80 | 1.00 | 1.00 |
| 6 | GPT-3.5-Turbo | 0.90 | 0.95 | 0.95 | 0.20 | 0.55 | 0.60 | 0.00 | 0.05 | 0.05 |
| 6 | GPT-4 | 1.00 | 1.00 | 1.00 | 0.90 | 0.95 | 0.95 | 0.60 | 0.90 | 1.00 |

Table 2: Preorder Traversal with LLMs

| Depth | Model | No Reduction | | | Single Step Reduction | | | Full Reduction Trace | | |
|---|---|---|---|---|---|---|---|---|---|---|
| | | Hit@1 | Hit@3 | Hit@5 | Hit@1 | Hit@3 | Hit@5 | Hit@1 | Hit@3 | Hit@5 |
| 3 | GPT-3.5-Turbo | 0.45 | 0.75 | 0.80 | 0.45 | 0.55 | 0.65 | 0.45 | 0.75 | 0.85 |
| 3 | GPT-4 | 0.70 | 0.95 | 1.00 | 0.80 | 0.90 | 0.95 | 0.85 | 1.00 | 1.00 |
| 4 | GPT-3.5-Turbo | 0.20 | 0.50 | 0.55 | 0.20 | 0.45 | 0.70 | 0.10 | 0.20 | 0.35 |
| 4 | GPT-4 | 0.75 | 0.85 | 0.90 | 1.00 | 1.00 | 1.00 | 0.50 | 0.80 | 0.90 |
| 5 | GPT-3.5-Turbo | 0.05 | 0.10 | 0.25 | 0.10 | 0.30 | 0.40 | 0.00 | 0.00 | 0.10 |
| 5 | GPT-4 | 0.50 | 0.90 | 0.95 | 0.60 | 0.90 | 0.90 | 0.10 | 0.25 | 0.45 |
| 6 | GPT-3.5-Turbo | 0.00 | 0.00 | 0.00 | 0.00 | 0.25 | 0.25 | 0.00 | 0.00 | 0.00 |
| 6 | GPT-4 | 0.15 | 0.35 | 0.40 | 0.55 | 0.75 | 0.75 | 0.00 | 0.20 | 0.20 |

Table 3: Inorder Traversal with LLMs

not handling the recursive cases properly, and, more predominantly, not being able to terminate the reduction process after mapping `X0` to `X1` as illustrated in Figure 23 (71 percent for GPT-4).

In analyzing the capabilities of the GPT-x models, it is evident that, while they can identify common patterns among presented examples to a certain degree, they lack the intrinsic ability to fully "comprehend" and execute the concept of structural recursion solely from these examples. This limitation is noteworthy, considering the models' demonstrated proficiency in a myriad of other computational tasks.

The insufficiency of these models in accurately performing step-wise reductions becomes particularly pronounced in the context of tree traversal experiments. In these experiments, the performance of LLMs exhibits a significant decline as the tree depth increases, correlating with a rise in the complexity and number of reduction steps required. Interestingly, both models observably achieve better performance when no in-context reduction is required. These LLMs appear to have have learned the specific concept of tree traversals from extensive pre-training and are observed to be capable of performing traversals even in zero-shot up to some accuracy, so inaccurate in-context reductions mislead the model to produce worse results.

```
Reduction Chain By GPT-4

S(X1 X1 X1 X1 X1 X1 X0 X0 X0 X0 01)
= X0 S(X1 X1 X1 X1 X1 X0 X0 X0 X0 01)
= ...(correct steps)
= X0 X0 X0 X0 X0 X0 S(X0 X0 X0 X0 01)
= X0 X0 X0 X0 X0 X0 X1 S(X0 X0 X0 01)   <- Error begins
= X0 X0 X0 X0 X0 X0 X1 X1 S(X0 X0 01)
= X0 X0 X0 X0 X0 X0 X1 X1 X1 S(X0 01)
= X0 X0 X0 X0 X0 X0 X1 X1 X1 X1 01
```

Figure 23: Non-stopping Error Made By GPT-4.

The degradation in performance during in-context reduction of the preorder traversal (Table 2) serves as a rudimentary benchmark for understanding how the models' accuracy declines in relation to increased task length. The fundamental challenge in preorder traversal is the sequential reading of numbers and peeling off of parentheses, which becomes progressively difficult as the sequence extends. However, the same simple trick seen in our toy models of copying node values for preorder traversals is again exploited by the LLMs.

More complex is the case of inorder traversal (Table 3), where a qualitative analysis reveals a higher propensity for errors in the intermediate reduction steps. In these instances, the LLMs frequently deviate from the correct reduction algorithm. A typical error involves the misinterpretation of the reduction rule:

```
inorder (Branch val l r) ⎯δβι→ (inorder l) ++ [v] ++ (inorder r)
```

to instead be:

```
inorder (Branch val l r) ⎯δβι→ [v] ++ (inorder l) ++ (inorder r)
```

though we denote this slightly differently in our prompting for convenience.

---

**Reduction Chain By GPT-4**

inorder[901 ( 907 LEAF ( 437 LEAF LEAF ) ) LEAF]
=inorder[907 LEAF ( 437 LEAF LEAF ) ] 901 inorder[LEAF]
**=inorder[ LEAF ( 437 LEAF LEAF ) ] 907 inorder[LEAF] 901** <- Error begins
... more steps

---

Figure 24: Example of mistake made by GPT-4 during in-order traversal.

This type of error underscores the challenges LLMs face in maintaining algorithmic consistency over extended sequences, particularly when the task requires adherence to a precise order of operations.

## 11 Discussion: Theory vs. Practice

One of the themes that permeated all of the experiments we ran, and our design of the framework itself, is the gap between theory and practice for learning to emulate the behavior of recursive functions. There is a long precedent of constructing models that behave exactly as expected. But what does that mean with respect to what can actually practically be learned? How can we observe this gap?

To revisit this gap in light of our empirical results, let us consider again the binary successor task. Following our framework's reduction semantics and viewing the constructor `S` as a function, a single-step reduction on the binary successor task breaks down into three cases, with each case effectively corresponding to a sub-string replacement:

1. `S ∘ X0` $\xrightarrow{\delta\beta\iota}$ `X1`
2. `S ∘ X1` $\xrightarrow{\delta\beta\iota}$ `X0 ∘ S`
3. `S 01` $\xrightarrow{\delta\beta\iota}$ `X0 01`

As expected, transformer models can *clearly* learn each individual reduction step of this simple function form examples. To determine this, we experimented with both 1-layer and 2-layer encoder-decoder transformers, and found that both kinds of transformer models indeed *can* solve such tasks with perfect accuracy (Table 4). This indicates the sufficiency of the models' capabilities to solve such sub-string recognition and replacement tasks in both orders.

But recursion is more than just reduction; it is also composition. If the model were to learn said composition organically, this would give rise to a $\mathcal{O}(l)$-layer transformer solution to compute binary successors where $l$ is the sequence length, assuming we also have some termination mechanism. This is bounded in size and quite

| Order | Model | $L_{max}^{Train} + 2$ to $+4$ | $L_{max}^{Train} + 4$ to $+6$ | $L_{max}^{Train} + 8$ to $+10$ | $L_{max}^{Train} + 18$ to $+20$ |
|-------|-------|------|------|------|------|
| Natural | 1-Layer | ✓ | ✓ | ✓ | ✓ |
|  | 2-Layer | ✓ | ✓ | ✓ | ✓ |
| Reverse | 1-Layer | ✓ | ✓ | ✓ | ✓ |
|  | 2-Layer | ✓ | ✓ | ✓ | ✓ |

Table 4: One-step reduction results for binary successor task. ✓denotes achieving $>99\%$ accuracy in each individual depth for all steps in a reduction in that test group.

inefficient, but it is possible—if it can actually be learned. But experimenting to see whether this happens organically becomes quite intractable for larger sequences.

The manual construction of such a solution—the theory—is also straightforward for this task. Transformers are shown to be constructed to achieve Turing completeness by way of repeatedly applying a transformer with a fixed depth. For this use case, one can construct a single-layer causal transformer that performs one-step reduction for the binary successor task. By stacking these layers and implementing a simple mechanism that checks the convergence of the algorithm (i.e., searching for the existence of the S symbol), such a model can compute the successor of a binary string in the constructor notation. For a fixed recursive depth $D$, this would take $D$ layers to construct. This is a bit less inefficient than length bounding, but it is still quite intractable to test empirically for high recursive depths.

There are also ways to construct by hand transformer models that perform the entire computation for the binary successor problem up to a *bounded* length, without necessarily being quite as intractable in terms of depth. These models could work by searching for consecutive X1s at the end and deciding whether to produce an X1 then copy the rest, or to produce X0 O1, based on what comes after the deepest X1. This is not quite what we saw learned in practice, but it does match our empirical findings that these models tend to learn shortcuts that are specific to particular sequence lengths or recursive depths.

The tree traversal is a much more difficult task. Building upon existing works on Dyck language generation and other constructions (Ebrahimi et al., 2020; Liu et al., 2022), we beleive that it should be straightforward to prove the existence of manually constructed models that perform one-step reduction on the task of tree traversal by adding layers that copy node values with pre-labeled depths of parentheses. This does not match what we observed in practice, and this gap warrants further exploration.

## 12   Related Work

**Understanding Transformers**   As a predominant class of deep learning model architectures, the transformer has shown promise in a wide range of tasks. Researchers have been attempting to understand the models via different approaches under different problem set-ups.

Methodology-wise, a line of research aims to characterize the models' capabilities via theoretical constructions (Bhattamishra et al., 2020b;a; Yun et al., 2020; Ebrahimi et al., 2020; Wei et al., 2023; Pérez et al., 2021; Liu et al., 2022; Merrill & Sabharwal, 2022; Pérez et al., 2021; Hahn, 2020). These studies effectively answer the questions revolving around the expressiveness of the transformer architecture by showing the existence of a model that satisfies the properties under study. Our work differs in methodology, as we adopt empirical methods in order to answer the question of whether the models can learn to emulate recursive functions from I/O examples, where the emphasis is the ability to mine recursive rules from data in practice. Another class of work takes a different approach where they seek to understand the functions of different components of a trained model via probing  (Hewitt & Manning, 2019; Belinkov, 2022), analyzing attention patterns  (Vig & Belinkov, 2019; Sun & Marasović, 2021), or linear layers (Geva et al., 2020).

Our analyses of the algorithms performed by our trained models were inspired by existing work in the relatively new field of mechanistic interpretability (Nanda et al., 2023; Chughtai et al., 2023) and include methods such as counterfactual patching (Meng et al., 2023), circuit tracing (Wang et al., 2023a), automatic circuit discovery  (Conmy et al., 2023), and component ablation. Mechanistic interpretability encodes the merits of the ASM concept since it assumes the learned model to be a program and attempts to understand

the model by understanding the program it represented at a certain level of abstraction. In our work, we use several of these techniques to reverse-engineer the critical components of the model and how they carry out the algorithm that solves the tasks in question.

One other key component of our framework is the introduction of structural recursion tasks formulated in a way that connects two paradigms (programming language and machine learning) and two key ingredients of programs (syntax and semantics). A stream of previous work (Delétang et al., 2022; Zhang et al., 2022a;c; Yao et al., 2021; Liu et al., 2023a) also took the approach of designing toy task to understand the computational ability or isolate a certain aspect of a class of deep learning models.

**Formal Theorem Proving and Program-Related Reasoning**   The tasks we choose are inspired by work in program and proof synthesis and repair (Gulwani et al., 2017; Lee & Cho, 2023; Osera & Zdancewic, 2015; Miltner et al., 2019; Ringer et al., 2021; Ringer, 2021; Magaud & Bertot, 2000; Chaudhuri et al., 2021), and are also an important class of functions for inductive logic programming (Cropper et al., 2021). Transformer-based large language models have brought significant progress to tasks related to programs (Chowdhery et al., 2022; Chen et al., 2021; Svyatkovskiy et al., 2020; Lu et al., 2021; Nijkamp et al., 2022; Xu et al., 2022; Zheng et al., 2023; Ahmad et al., 2021; Chakraborty et al., 2022; Wang et al., 2021; 2023b; Li et al., 2022a; 2023; Xu et al., 2023; Fried et al., 2022; Allal et al., 2023; OpenAI, 2023a;b; Thoppilan et al., 2022) and proofs (Polu & Sutskever, 2020; Han et al., 2022; First et al., 2023). Still, current tools struggle to emulate function behavior (Gu et al., 2024) without prompt engineering techniques like chain of thought prompting (Wei et al., 2022) or scratchpad reasoning (Nye et al., 2021).

On the other hand, execution turns out to be an essential element for better program synthesis with LLMs (Olausson et al., 2023; Le et al., 2022; Pan et al., 2023; Zhang et al., 2023). A series of code language models specializing in execution-related tasks have been introduced to tackle specific problems in software engineering (Souza & Pradel, 2023; Pei et al., 2023; Ding et al., 2023; Liu et al., 2023b).

**Formal Languages and Deep Learning Models**   Formal languages exhibit clear structures, encode certain rules, and can be easily generated; thus, they are particularly useful in characterizing neural networks' representational capacities. They have long been used as a tool to study the learning abilities of machine learning systems, especially neural architectures (Kleene, 1951; Casey, 1996; Smith & Zipser, 1989).

More recently, the rapid advancement of deep learning technologies has made possible different applications of neural networks, such as modeling natural and programming languages. It has been argued that the ability to simulate formal language is correlated with the ability to capture natural language grammar and solve complex problems (Merrill, 2023; Strobl et al., 2023). In addition, compared with real-world tasks that often involve additional complexities, formal language modeling offers a controlled setting where we can abstract away the irrelevant factors to better benchmark the capabilities and identify the limitations of the models (van der Poel et al., 2023; Liu et al., 2022; 2023a).

**Generalization Capabilities of Transformers** Numerous studies have meticulously examined the limitations of transformers in out-of-distribution generalization through both algorithmic and formal tasks (Delétang et al., 2022; Anil et al., 2022; Zhang et al., 2022c; Dziri et al., 2023; Veličković et al., 2022; Liu et al., 2022). Generalization can be assessed along different axes: length generalization, which pertains to the model's capacity to handle longer sequences than those encountered during training; and algorithmic generalization, where the model must execute more complex procedures.

There have been many attempts to improve model's generalization to longer sequences through positional encoding and data format (Shaw et al., 2018; Dai et al., 2019; Ruoss et al., 2023; Press et al., 2022; Raffel et al., 2020a; Zhou et al., 2024). On the other hand, algorithmic generalization has also been investigated from architecture, training and task perspectives (Ouellette et al., 2024; Kamali & Kordjamshidi, 2023; Weiss et al., 2021; Zhou et al., 2023).

## 13   Conclusions, Limitations, and Future Work

In this paper, we proposed a comprehensive framework to study the sequence models' behaviors in *learning to emulate* structural recursion. The framework we proposed takes care of both pillars of the field of pro-

gramming language domain: syntax and semantics. Syntax-wise, we introduced a new sequential encoding that nicely transforms recursive data structures into sequences. Semantics-wise, we connected the notion of structural recursion to a broad range of ML-based sequence models that encompass different settings—both gradient-based training and in-context learning.

To cover two representative classes of recursion—one that has only recursive sub-case at each reduction step (i.e., calls the reduction linearly) which is naturally sequential, and another that has multiple sub-cases at each step that forms a tree structure—we instantiated the framework to two simple yet structurally interesting tasks that are themselves meaningful to formal reasoning and programming: (1) **binary successor**, and (2) **tree traversal**.

We focused on studying the recursion problem-solving capabilities of the predominant transformer-based models. Guided by the proposed framework, we performed empirical studies and careful inspections into the model's learned mechanism. Our findings suggest that the models trained to approximate recursive computations learn programs that are non-recursive and thus fall for edge cases or are limited in capacity to generalize across different depths of recursion. Our analysis and mitigation efforts show that data distribution and the model configuration could directly hinder the learned model's performance. However, we hypothesize the fundamental cause is that the model has not been optimized to represent recursive patterns well enough. Our in-context learning experiments revealed the lack of ability for LLMs to "think recursively" as it tends to mine non-recursive rules from data. Also, the models lack accuracy when performing step-wise reduction, where they may get lost in the middle or fail to apply the stopping condition.

Recursion as a universal concept in language, computer science, and mathematics, is an inevitable challenge for any system to perform non-trivial problem-solving. Observing the lack of transformer-based models' abilities to capture structurally recursive patterns despite their impressive capabilities, we highlight the need to re-examine the extent to which recursive problem-solving capabilities have been effectively injected into transformer-based models through the existing training paradigm and whether&how it can be injected.

**Future works**  For future work, we are considering two directions: addressing the lack of success in recursion with current models and exploring their general ability to emulate program execution. Additionally, we aim to design deep learning architectures that better learn algorithmic patterns. We hope this work will serve as a foundation for empirical studies on the computational capabilities of deep learning models.

On the other hand, we aim to enhance task decomposition, particularly within programmatic reasoning setups. Recent studies (Wen et al., 2024; Drozdov et al., 2022; Yang et al., 2024; Shi et al., 2024) have explored task decomposition in programming-related contexts, highlighting its advantages. We envision a synergistic approach integrating these insights with emerging techniques such as reinforcement learning or preference learning (Mitra et al., 2024; Wang et al., 2024), along with curriculum learning methods that start from foundational algorithmic structures.

**Acknowledgements**  This research was developed with funding from the Defense Advanced Research Projects Agency. The views, opinions, and/or findings expressed are those of the author(s) and should not be interpreted as representing the official views or policies of the Department of Defense or the U.S. Government.

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

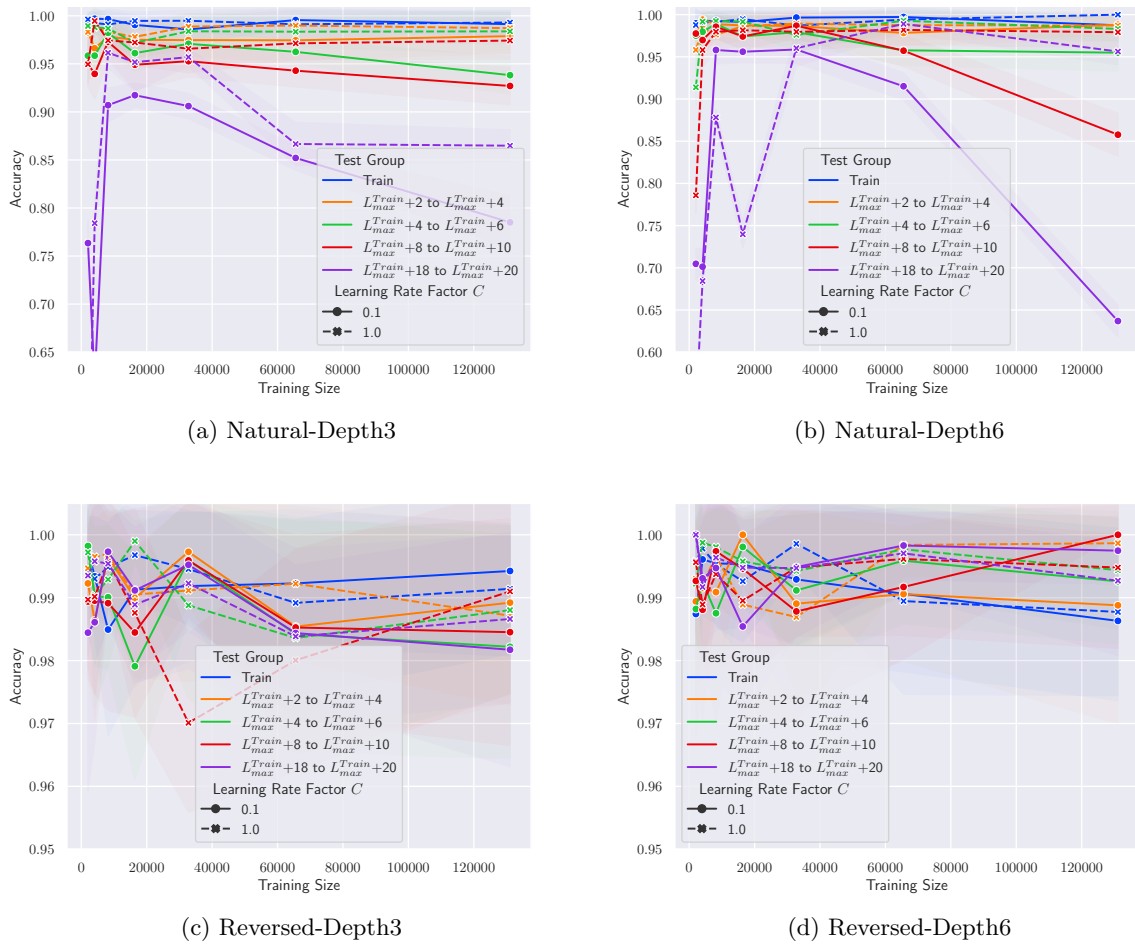

(a) Natural-Depth3        (b) Natural-Depth6

(c) Reversed-Depth3       (d) Reversed-Depth6

Figure 25: Model performance when trained on data with more constrained recursion depths.

# A    Depth-Restricted Training

One of the important aspects of generalization in a recursive problem is generalization across depths. We experimented with depth-restricted training, controlling the maximum depth the model saw during training. Notably, the generalization performance between large and small learning rates exhibits a significant difference in natural order as shown in Figures 25a and 25b, whereas the same phenomenon did not occur in reversed order.

# B    Extrapolation of Different Model Architectures

We were initially not sure whether to evaluate the tasks of interest on an encoder-only, decoder-only, or encoder-decoder transformer model. To decide, we ran preliminary experiments to evaluate these three architectures under our setup for a related toy experiment and on one of our two tasks.

Our toy experiment focused on string reversal. We trained models on strings ranging from 10 to 37 and tested up to length 50. We used three different architectures: a 2-layer encoder-decoder transformer model, a 4-layer encoder-only model, and a 4-layer decoder-only model. We trained until we achieved perfect accuracy on both the training and in-domain validation datasets. We employed the same "random-padding" strategy as in the main experiment.

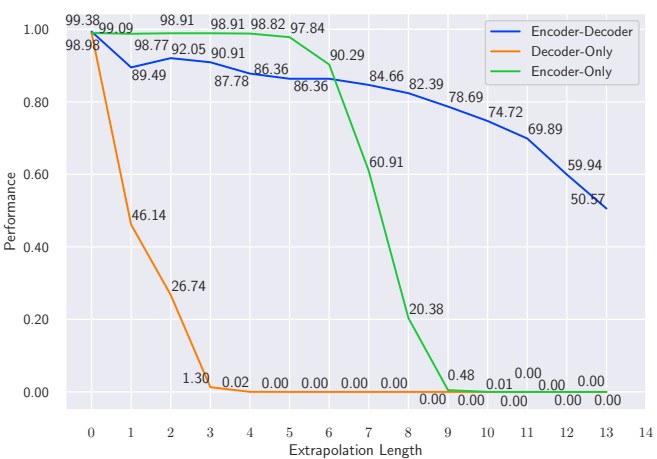

Figure 26: Performance of Extrapolation Study

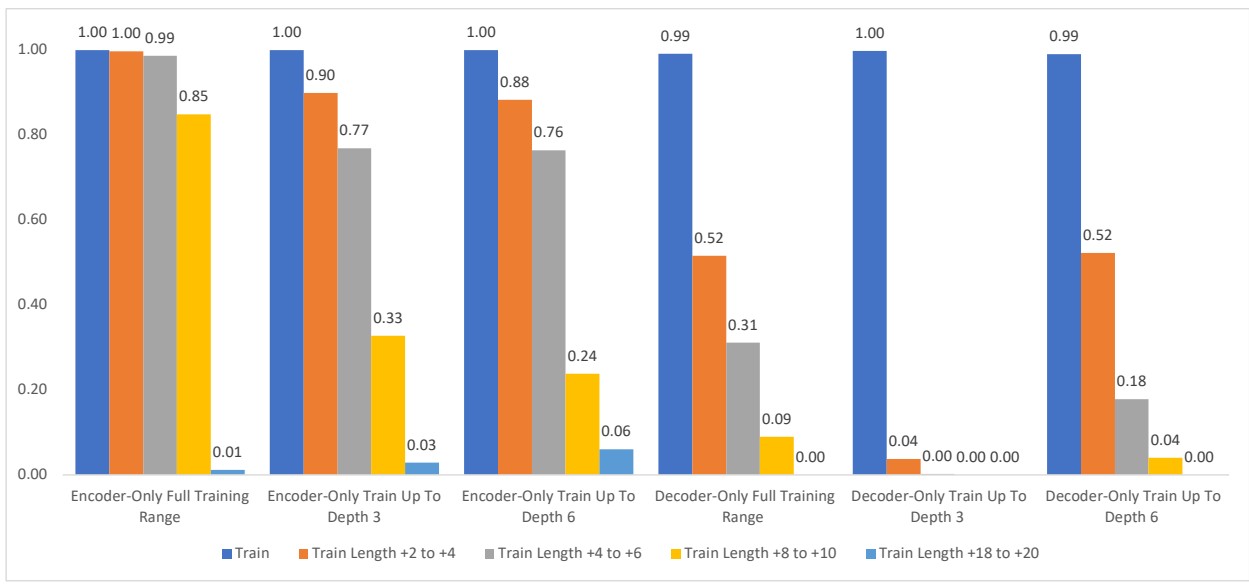

Figure 27: Accuracy of encoder-only and decoder-only transformers on natural order binary successor task.

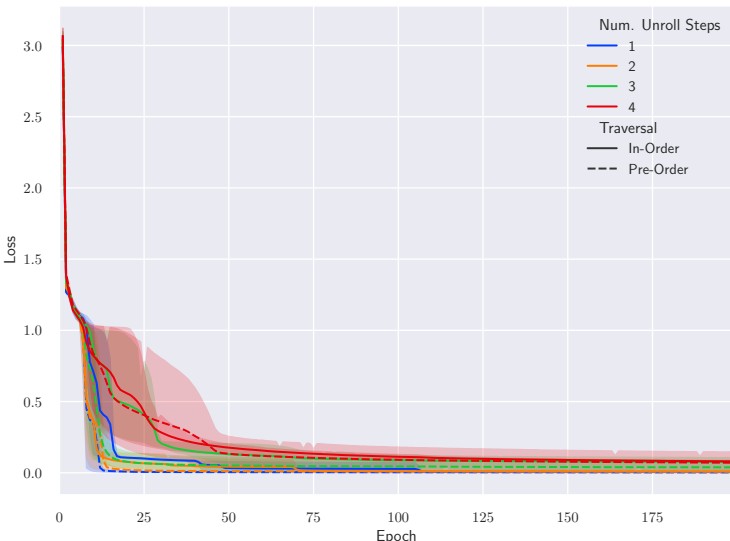

Figure 28: Tree Traversal Loss vs. Epochs

The encoder-decoder model exhibited the best extrapolation performance among the three architectures (Figure 26). Both the encoder-only and decoder-only models experienced a rapid decline in performance as the extrapolation length increased. Interestingly, we also observed that the model failed to extrapolate when trained on a fixed-length "learnable token" setup, even though we ensured that the positional embeddings were seen during training. Moreover, the model also struggled to extrapolate when padding was not applied.

The results of the encoder-only and decoder-only transformers on the natural order binary successor task (each with four layers) are presented in Figure 27.

## C   Sketch of Proof: Single layer transformer can solve one-step reduction of binary successor task.

Let us assume, for simplicity, 4 types of tokens:
S, 01, X0, X1.

We assume the token embeddings to be one-hot
$Emb([BOS]) = [0, 0, 0, 0, 0, 0, 0, 0]$
$Emb(S) = [1, 0, 0, 0, 0, 0, 0, 0]$
$Emb(X0) = [0, 1, 0, 0, 0, 0, 0, 0]$
$Emb(X1) = [0, 0, 1, 0, 0, 0, 0, 0]$
$Emb(01) = [0, 0, 0, 1, 0, 0, 0, 0]$
.

We assume the $W_Q$ matrix to extract the positional encoding of the previous token.
Assuming the positional encoding is one-hot encoding of length $L$, the $W_Q$ should be able to obtain the one-hot encodings of the preceding positions of each location (A permutation matrix). $W_K$ should output $0, 1, ..., L$ (i.e. identity matrix). $W_V$ should return token embedding of that token.
Assuming causal mask and residual connection after Softmax: attention output $AttnOut = Softmax(QK^T)V + X$ where $X$ is the input to this layer. The goal is to encode the tokens preceding each token into the second four slots of the embedding vector. $[BOS]$ token, with embedding vector being $\mathbf{0}$, will be written as 0 to it succeeding token's output vector. For simplicity, we write each entry in the output of this layer $[x, y]$ where $x$ is the token embedding of this entry, $y$ is that prior to this entry.

The MLP is supposed to perform a simple non-linear operation on the output such that

```
Procedure BinarySuccessorNaturalOrder
  Input: BinarySequence b
  Output: BinarySequence b'
  HyperParameters: maxDecodingLength
  Initialize decodingCounter=1,isInRecursion=False
  Initialize BinarySequence b' = []
  recursionStartPosition,sequenceLastPosition = Model.Encoder.recognizeImportantPositions(b)
  while decodingCounter < maxDecodingLength do
    startRecursionFlag = Model.Decoder.checkIsRecusrionStart(\
    Model.Decoder.positionalEncoding(decodingCounter),\
    recursionStartPosition)
    if not startRecursionFlag:
      if startRecursionFlag = Model.Decoder.checkIsLastBit(\
        Model.Decoder.positionalEncoding(decodingCounter),\
        sequenceLastPosition):
        bit = Model.Decoder.generateX1()
      else:
        bit = Model.Decoder.copy(b[decodingCounter])
    else:
      if not isInRecursion:
        isInRecursion=True
        bit = Model.Decoder.generateX1()
      else:
        bit = Model.Decoder.generateX0()
  b'.append(bit)
  if Model.Decoder.positionalEncoding(decodingCounter) == sequenceLastPosition:
    b'.append("<\s>")
    return b'
  else:
    decodingCounter += 1
```

Figure 29: Reconstructed Algorithm: Natural Order

- maps the token embeddings of $[S, X1]$ into the embeddings of $[X1, S]$

- maps the token embeddings of $[S, X0]$ into that of $[-, X1]$

- maps the token embeddings of $[S, 01]$ into that of $[X1, 01]$

- maps the token embeddings of any other pairs to themselves.

We can implement an MLP layer such that it checks whether each pattern above occurs, and an additional condition of whether $S$ ever appears in the input sequence. The first part can be done easily by multiplying each embedding by a matrix $M$ with row vectors denoting the pattern, and applying non-linearity on $Mv-1$. To check if $S$ ever appeared in the sequence to determine whether or not to halt, one could simply check the first row of the output $O$ is positive by applying non-linearity.

# D    Reconstructed Algorithms

## D.1    Algorithm Reconstruction for Binary Successor

We summarized our results for the binary successor task in Section 7. Here, we provide pseudocode for the reconstructed algorithms in both the natural (Figure 29) and reverse(Figure 30) orders.

## D.2    Algorithm Reconstruction for Tree Traversals

Using the technique we introduced in 8.1, we discover the algorithms learned by the model during tree traversals. The results indicate that one head prioritizes future EMPTY and parentheses tokens, while the other focuses on the token that requires copying from the original sequence. This provides insight into the attention patterns exhibited by the model at different processing stages, highlighting the key features that the model attends to.

The learned transformer model employs a set of logical rules and functions facilitated by the attention mechanism for tasks such as copying tree node values or producing special tokens. It uses various types

```
Procedure BinarySuccessorReverseOrder
  Input: BinarySequence b
  Output: BinarySequence b'
  HyperParameters: maxDecodingLength
  Initialize decodingCounter=1,isInCopyMode=False
  Initialize BinarySequence b' = []
  while decodingCounter < maxDecodingLength do:
    inCopyMode = Model.Decoder.hasGeneratedX1(b')
    if inCopyMode:
      bit = b[decodingCounter]
    else:
        if Model.crossAttentionCheckIfFirstX1(Model.Encoder.findFirstX1(b),\
          Model.Decoder.InternalState(b')):
          bit = Model.Decoder.GenerateX1()
        else:
          bit = Model.Decoder.GenerateX0()
    b'.append(bit)
    decodingCounter += 1
  return b'
```

Figure 30: Reconstructed Algorithm: Reverse Order

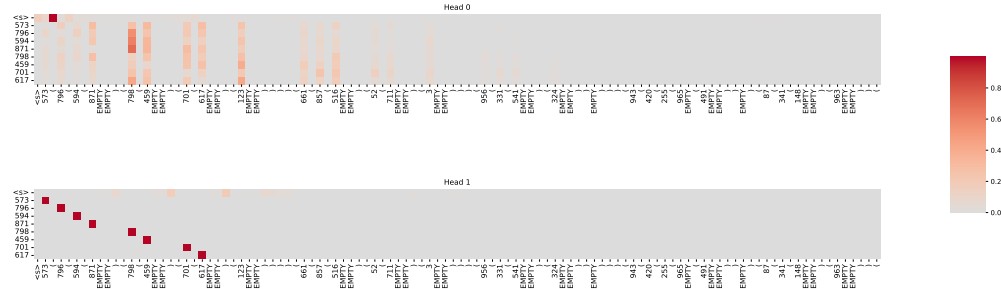

(a) Layer 1 cross-attention. The model attends only to nodes, not to parentheses or EMPTY tokens.

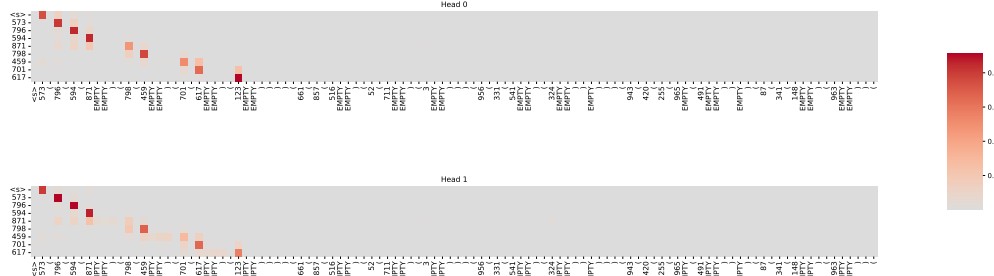

(b) Layer 2 cross-attention. Attention is consolidated on the next node to be copied, skipping all non-node items.

Figure 31: Cross-attention on the pre-order full traversal task.

of attention to keep track of important token locations for either logical control flow or invocation of the mechanisms to produce the next token in the output sequence. In the task of in-order twice reduction, for instance, the model utilizes its attention mechanism to identify the appropriate reduction level and pinpoint the node whose values will be copied to the current output slot, as illustrated in Figure 32b. The model combines multiple logical conditions to discern the boundary separating the remaining sub-trees and the layers set for unrolling. To generate the `"UNROLL["` statement in pre-order reduction, the model's cross-attention is focused on three key positions: 1) the parenthesis immediately preceding the node where the `"UNROLL["` token will be inserted, 2) the subsequent opening parenthesis, and 3) the final closing parenthesis of the subsequence that will be copied into the `"UNROLL[ ]"` statement.

**In-order Partial Reduction Traversal (Example: Reduce Twice, Depth 2-3)**

1. The network always starts by inserting an "UNROLL[" symbol.

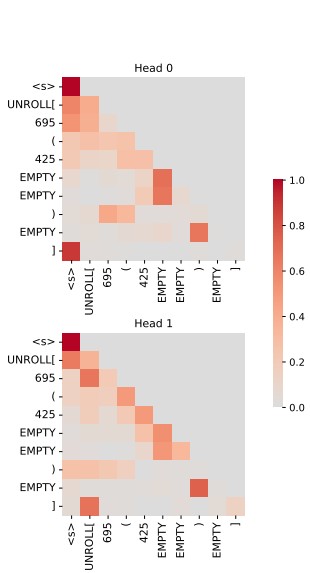

(a) Layer 1 decoder self-attention during in-order partial reduction at a higher reduction depth.

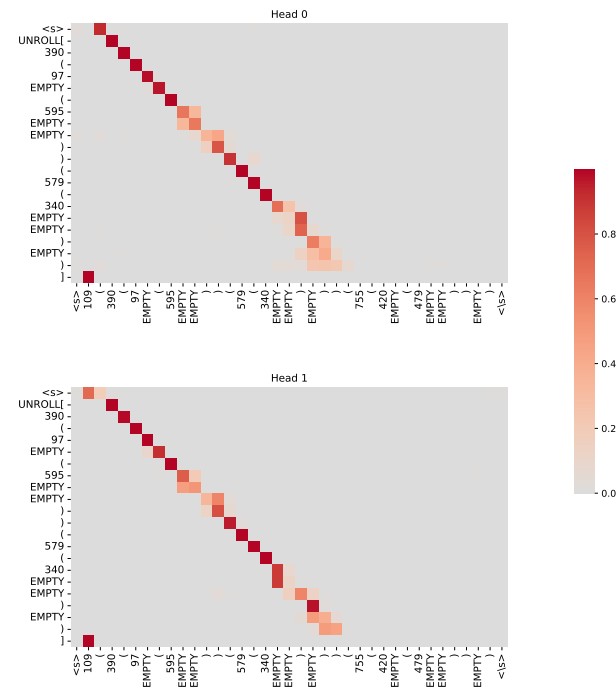

(b) Layer 1 cross-attention at the end of the left sub-tree to be unrolled.

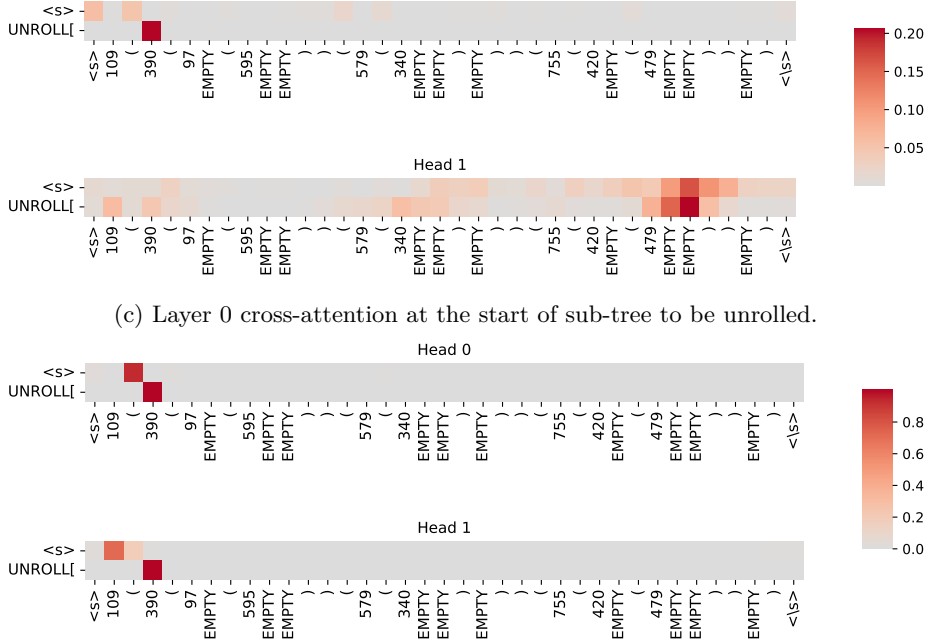

(c) Layer 0 cross-attention at the start of sub-tree to be unrolled.

(d) Layer 1 cross-attention at the start of sub-tree to be unrolled.

Figure 32: Cross-attention during inorder traversal task.

2. Cross-attention attends to the token after peeling off two (since it is a twice-reduction) layers of parentheses and writes this token into the final token position.

3. This process continues with the model attending to target tokens and parentheses in sequence until the first parenthesis is closed. It identifies the current depth by jointly attending to the open parentheses on the input side and the previously closed parentheses

4. The model inserts a closing "]" bracket. The left subtree is now complete.

5. The model attends to the token immediately preceding the closed first parenthesis and writes this token out.

6. The model then performs the same copy operation with the right subtree.

For higher depths, the model performs essentially the same operation but additionally uses decoder self-attention to attend to the beginning and end of the relevant bracket level as per Figure 32a. The significance of this attention pattern seems to suggest that it helps the network track which parent node to copy over. For shallower reductions, the model only needs to copy over the root node at the beginning of the encoder sequence, so it does not need to track multiple parent nodes; however, for deeper trees, the model needs some indication of where the appropriate parent node can be found. Potentially, the model could use the start of the "UNROLL[ ]" sequence as a way to point to the token immediately *after* the appropriate parent node, which could then be used as a query to find the node prior to it in the original encoder sequence.

**Pre-order Partial Reduction Traversal (Example: Reduce Twice, Depth 2-3)**

1. The model begins by copying node numbers without parentheses.

2. Depending on the depth, the model is trained to output an "UNROLL[" token once two reductions have occurred. When writing out this token, cross-attention patterns of the key heads indicate that the model is attending to A). the parenthesis immediately prior to the node after which the "UNROLL[" token is to be inserted, and B). the following opening parenthesis, and C). the final closing parenthesis of the subsequence that will be copied into the "UNROLL[...]" statement. This seems to suggest that the model is keeping track of the parenthetical depth when deciding where to insert the "UNROLL[...]" wrapper.

3. In parts of the output sequence into which an EMPTY token would normally be inserted with a simple copy operation, decoder self-attention is more significant to the final logit difference. Looking at attention patterns once again, we see that the model is attending to the preceding UNROLL[ token, if present, and is likely using it as a basis for whether to output this token or not. (Subsequences inside the wrapper should contain "EMPTY" tokens, while those outside should not.)

4. The model completes the sequence with the end token ("<\s>"). It is less clear what the model is doing here, but the most significant contributing heads attend to a combination of the final brackets and EMPTY tokens as well as the <\s> token of the original encoder output.

## D.3   Separation of Tasks in Attention Heads

The analysis of attention patterns reveals important insights. In the in-order traversal task, Layer 1 cross-attention heads show distinct behavior: one head focuses on forward parenthesis, brackets, and EMPTY tokens, while the other attends to encoder sequence tokens linearly (Figure **??**). At Layer 2 (Figure 32d), both heads correctly attend to the token to be copied, suggesting they write out the next correct token based on Layer 0's output. Brackets and symbols play a crucial role in signaling behavior changes, particularly in steps requiring the insertion of non-consecutive symbols. Decoder self-attention heads attend to previous UNROLL[ symbols when determining the inclusion or omission of EMPTY tokens. For in-order traversal, higher depths involve attending to bracket statements for copying root nodes, unlike binary trees with a single root node. This behavior may be utilized by decoder cross-attention heads to copy the first node within the UNROLL statement.

Our analysis indicates that the individual encoder self-attention heads do not appear to play a significant independent role in the transformer model's performance. This is demonstrated by our counterfactual patching experiments, which resulted in minimal changes to the network's predictions. On the other hand, decoder cross-attention appears to be a crucial component of the transformer circuit. As the sequence progresses, however, decoder self-attention becomes increasingly important and the model takes into account the structure of the previously-generated content (especially at higher depths, where keeping track of multiple levels of parent nodes becomes vital).

### D.4   Pseudocode for dataset generation

```python
from typing import List

def S(input_binary: List[str]) -> List[str]:
    if input_binary == ['01']: # Base case
        return ['X0', '01']
    elif input_binary[0] == ['X0']: # Non-recursive case
        return ['X1'] + input_binary[1:]
    elif input_binary[0] == ['X1']: # Recursive case
        return ['X0'] + S(input_binary[1:])

    from typing import List, Union

class TreeNode:
    def __init__(self, val=0, left=None, right=None):
        self.val = val
        self.left = left
        self.right = right

def print_tree(root: Union[TreeNode, None]) -> Union[List[str], None]:
    # Base case: If the root is None, return None to represent an empty subtree.
    if root is None:
        return None

    # Recursively print the left subtree and the right subtree.
    left_subtree = print_tree(root.left)
    right_subtree = print_tree(root.right)

    # Construct branches for the left and right subtrees.
    if left_subtree is None:
        left_branch = ['LEAF']
    else:
        left_branch = ['('] + left_subtree + [')']

    if right_subtree is None:
        right_branch = ['LEAF']
    else:
        right_branch = ['('] + right_subtree + [')']

    # Construct the representation of the current tree node and its subtrees.
    tree_representation = [str(root.val)] + left_branch + right_branch
    return tree_representation
```

## E   Additional Visualizations

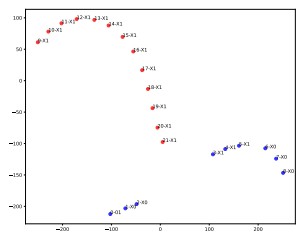

(a) T-SNE plot for the encoder embedding.

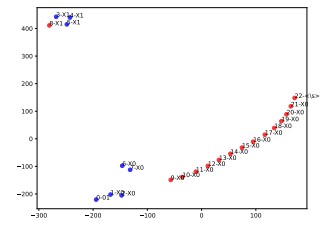

(b) T-SNE plot for the decoder layer 0 self-attention output (before cross-attention).

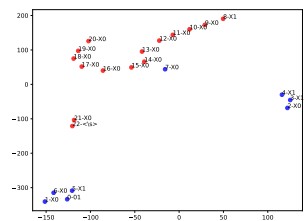

(c) T-SNE plot for the decoder layer 1 self-attention output.

Figure 33: Visualization of T-SNE plots.

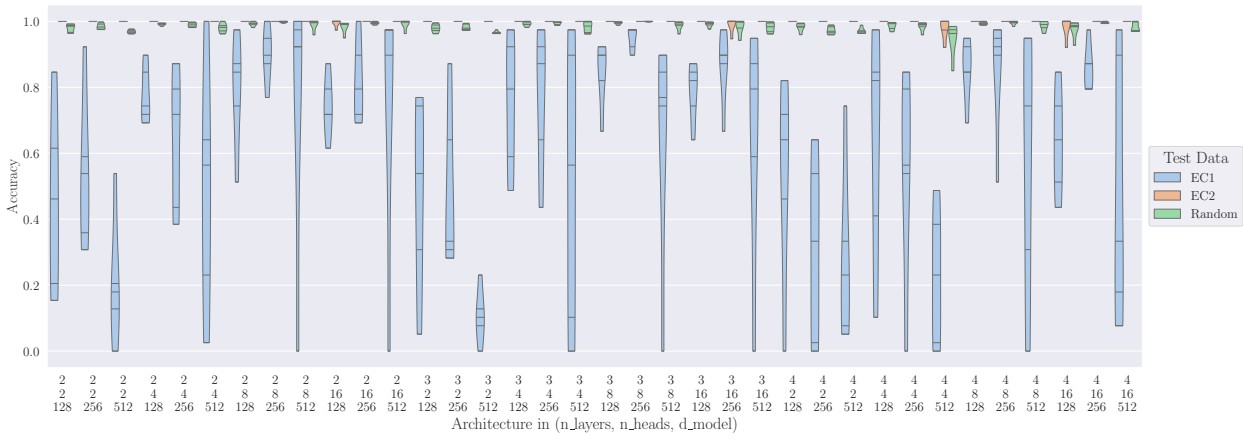

(a) Results on up-scaling models on binary-successor reverse order.

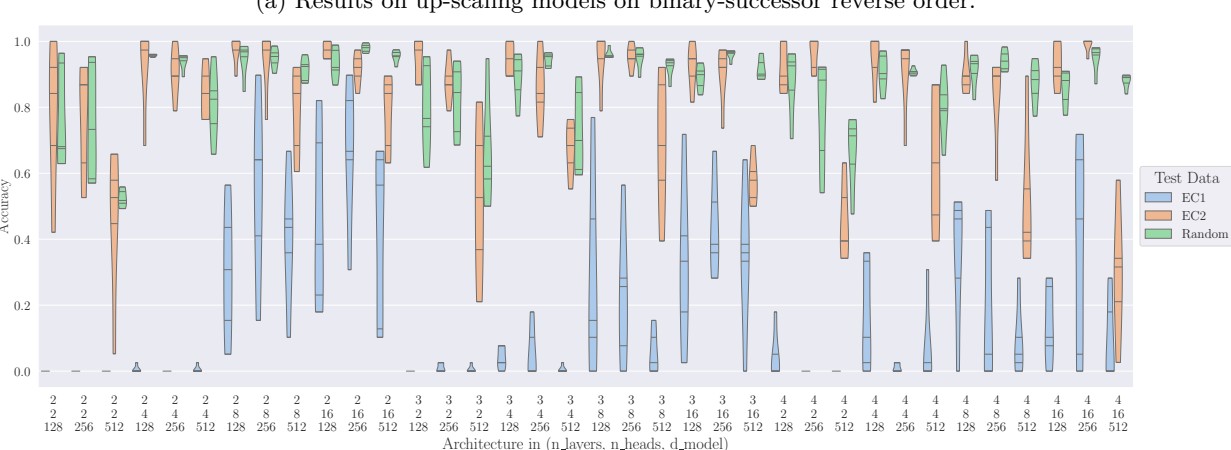

(b) Results on up-scaling models on binary-successor natural order.

Figure 34: Up-scaling models on binary successor task. We observe slight benefits from increasing the model size (i.e. hidden dimension, number of heads and layers.)

