# OpenReview forum: "Transformer-Based Models Are Not Yet Perfect At Learning to Emulate Structural Recursion"
_TMLR — Accepted by TMLR_

### Review · Reviewer_QYN4 · 2024-04-21

**Summary Of Contributions:**

The paper belongs to a class of recent attempts to investigate the learning capability of transformers from the lens of formal grammar. It focuses on representing and reasoning about structural recursion - that is, recursion on data structures. It also reports on empirical investigation of learned models’ algorithmic behaviors on structurally recursive tasks. In particular, the paper studies two tasks: binary succession and tree traversal.

**Audience:**

Yes

**Broader Impact Concerns:**

There are no broader impact concern with this paper.

**Claims And Evidence:**

No

**Requested Changes:**

The paper's title, abstract and claims should be focussed to the actual investigation in the paper. Some of the unsupported claims in the paper can be dropped or provided with more evidence.

There is some repetition in the paper - defining and applying the framework can be merged. The discussion on representation of "syntax" and "semantics" can be consolidated into a single section. Similarly, the fact that only two tasks are being considered can be stated upfront and does not need to be repeated multiple times.

The discussion on the benefit of generalizability of inductive types is well-known in formal methods and several systems such as PVS, Coq, Lean have implemented these (in some cases for three decades). It seems to be orthogonal to the contributions of the paper. Similarly, the discussion on semantics and ASM is standard in any PL textbook and just a reference would suffice.

For "in-context learning", it is widely known that the models are sensitive to prompts - what measures were taken to ensure that the used examples and prompts were effective?

**Strengths And Weaknesses:**

* It is unclear what is meant by "framework" in this paper. It is not a new system or algorithm. It appears the authors are calling their investigation/study a framework - but frameworks are reusable and to some extent widely applicable. Here, the study was limited to just two structurally recursive tasks: (a) learning the binary successor function and (b) learning tree traversals. The actual claims in the paper title, abstract and introduction are much broader starting from discussions on program synthesis, theorem proving, textual occurrences of recursion, etc. while the technical contribution is quite narrow.

* There are some broad statements in the paper that are not well-supported by analytical reasoning or empirical data. They are alos not warranted for the main topic of the paper and could be dropped. "Transformer models are not architecturally recursive—they do not call themselves." While this is true superficially, can we make such a categorical declaration ruling out the possibility that the attention learned within transformers are implementing recursion - particularly, with the new moving average attention mechanisms that allow mimicking memory across contexts. Also, when reporting hypothesis from the experiments, the claims are left rather soft such as " the transformer models we have trained do not strictly implement the kinds of state transitions one might associate with an RASM" - it would be good if the paper put forward some concrete observations even if they are conditioned rather than conclude with soft conjectures.

* Leaving aside a rehash of standard literature in formal methods and PL, the main contribution of the paper is in Section 5 wherein experiments were conducted on binary succession and tree traversal datasets. Section 7.4 on varying LR is interesting and could be expanded with explanation of the observations.

* The observation that transformer models can perform full preorder traversals on unseen trees, but fail completely for inorder traversals is very interesting and could have wider applicability.

---

> ### Author Response · Authors · 2024-04-30
> **Response Part 1 of 2**
>
> We thank the reviewer for their constructive suggestions and comments! We will make sure to address the issues pointed out by the reviewer during revision. We will consider the structural changes suggested by the reviewer to improve the conciseness, for example, by consolidating the discussions on syntax and semantics.
> We address the reviewer’s questions below.
>
> # Framework
>
> What we mean by “framework” in this paper is a conceptual framework. That is, a systematic way of (a) representing the syntax and (b) understanding the semantics of terminating structurally decreasing recursive functions, all in a way that is conducive to understanding the behavior of transformer models trained to approximate the behavior of these functions. The reusability and wide applicability of the framework comes from the fact that it can represent any terminating structurally recursive function. This is due to (a) the fact that induction and structurally decreasing recursive functions are equivalent, and (b) the fact that the reduction semantics are as general as the language at the core of Coq and Lean.
>
> The reviewer is of course correct that “the discussion on the benefit of generalizability of inductive types is well-known in formal methods,” which is why we cited existing work establishing the generalizability of the representation and of the reduction semantics. Our contribution was using the insight of this generalizability to build a powerful conceptual framework for understanding the behavior of transformer models trained to represent this class of functions. We will clarify all of this in the next revision.
>
> Regarding the choice of tasks, indeed we chose them to deliberately be fairly simple, in order to make it easier to investigate the behavior of transformer models on those tasks. At the same time, we chose them to be representative of two different recursive structures: (a) a single recursive case, and (b) a tree structure with multiple recursive cases. Though we stress that these are meant to serve as simple but interesting examples instantiating our broader framework, we do also discuss in the paper why these particular tasks are useful more broadly.
> # Concern about claims made in the paper
> We thank the reviewer for bringing up this concern. We will go through these claims and ensure they are either supported or rephrased/eliminated appropriately in the revision.
>
> We also would like to clarify with the reviewer some statements we made.
> Regarding the architectural non-recursiveness of Transformers: This is not the conclusion we draw from our experiments but instead the motivation of our empirical investigations. By saying transformers are architecturally non-recursive, we mean that transformer architecture was not explicitly designed to perform self-calling, thus there is no straight-forward answer to the question of whether transformer models can simulate a recursive algorithm. This is precisely the reason why we conducted experiments to better understand if a transformer model can learn a recursive function from input-output pairs. We will revise the phrasing to make this clearer.
>
>
> The discrepancy between model behavior and that of an RASM: We thank the reviewer for the seek of a clearer argument on RASM. It is a necessary condition for a model that implements an RASM to be able to encode the depth of recursion. However, our probing experiments show that the model did not capture such information in its learned representations. We therefore contend that standard sequence-to-sequence training practices tend not to induce behaviors of RASM
>
> # Interesting observations on LR and various traversals
>
> We are excited that the reviewer finds this part of the study interesting, and we share opinions with the reviewers on this!
> Indeed, the learning rate is observed to impact the model’s learned algorithm although converging to near-zero losses in all cases. As we presented in the paper, varying learning rates lead to different accuracies in models’ execution of the algorithm it learned, thus leading to different accuracies in computing the successor of binary numbers. This could be a very interesting direction to look into in the future as well!
>
> Additionally, we found it interesting that although the algorithms for in-order and preorder traversals are so similar in difficulty, they lead to completely different behaviors! We are also excited to hear more thoughts from the reviewers on this aspect!

---

> > ### Author Response · Authors · 2024-04-30
> > **Response Part 2 of 2**
> >
> > # In-context learning
> > GPT-3.5 and GPT-4 models are instruction-tuned to optimize their ability to understand and follow instructions of various forms.
> > Yet, we still followed the best practices in prompting these language models to ensure valid experimental results.
> > The format of the prompt followed the commonly adopted prompting practice. It is of the structure below:
> >
> > ```
> > ### Task Description
> > {Description of the task}
> > ### Examples
> > {Examples}
> > ### Formatting Requirements
> > {Formatting Requirements}
> > ### Input
> > {The input sequence}
> > ```
> >
> > As a sanity check, at the stage of prompt engineering we ensured the models can understand the task and formatting requirements specified in the prompt, and can produce correct results on simple cases.
> > As for in-context examples, one question of our interest is whether the LLM can summarize the rules to compute binary successor (which involves recursion) from a) consecutive numbers starting from base case and b) randomly sampled numbers.
> >
> > For tree traversals, we controlled the examples to be the same across all experiments.

---

### Review · Reviewer_APjT · 2024-04-22

**Summary Of Contributions:**

The paper proposes a framework for understanding structural recursion that separates syntax and semantics.
- For syntax, the paper proposes to describe it with an inductive type, which consist of base cases and inductive bases. Such description helps to separate syntax (i.e. structure) from semantics (i.e. meaning) while being sufficiently general. For example, it can capture programming languages.
- For semantics, the paper provides two choices: 1) stepwise reduction, following the 3 types of rules specified in $\delta\beta\iota$-reduction, and 2) abstract state machine (ASM) which provides a more general definition that can capture various structures encountered in math and programming.

The paper also conducts empirical investigation on Transformers' behaviors on 2 structurally recursive tasks, namely, learning the binary successor function and tree traversals. The paper includes interpretability analyses to understand the algorithms implemented by the learned models. The results suggest that Transformers do not necessarily follow the state transitions associated with an ASM, since Transformers may depend on spurious information such as the position. While the experiments mainly focus on models trained from scratched, there are also results on finetuned models and in-context learning, which find similar inability of learning recursive structures.

**Audience:**

Yes

**Broader Impact Concerns:**

There is no direct ethical or societal implication of the current work.

**Claims And Evidence:**

Yes

**Requested Changes:**

Please help address the clarifications and writing suggestions mentioned above.

**Strengths And Weaknesses:**

**Strengths**: The paper provides both a framework for studying structural recursion, and also empirical investigations on the practical considerations of trained models.

**Questions and comments**:
- Could you explain why ASM is considered as a part of semantics rather than syntax (structure)? If we think of replacing natural numbers with dinosaurs as replacing semantics while preserving syntax, isn't this also allowed in ASM?
- the phrase "sequential encoding" may cause potential confusion, if we think of Transformer as being more parallel than sequential as compared to, say, RNNs. Encoding may also be confused to encodings (e.g. hidden variables / outputs) of a neural network.
    - Moreover, currently "sequential encoding" seems to be a special concept proposed in this work, but it is actually not as it is how the syntax for e.g. programming languages are described (as also mentioned in the paper).
- task decomposition: this idea is related to chain of thought (CoT), where providing intermediate steps help with training or inference, but too many intermediate steps can be computationally inefficient. Hence an important question is to determine the type and granularity of intermediate supervisions. For instance, there could be some middle ground between the 2 cases in Sec 4.1.2, or potentially other (unintuitive) ways to decompose a recursive structure. Could you share your thoughts on this please?
- A related question to the above is how to understand errors in learning compositions of rules: how can we understand the errors in end-to-end computation, if there are ways to decompose the task other than following the reductions specified by the grammar?
- I don't necessarily agree with the comment in Sec 4.2.4 that mechanistic interpretability is an exercise of ASM-style analysis: mechanistic interpretability also involves discovering the algorithm implemented by a trained Transformer, which, although being (unavoidably) limited to the set of operations a person could think of, is more general than following the set of rules allowed by a given (but unknown) ASM. Please let me know if I misunderstand something.
- Sec 7.2: should I think of the reliance on positional encoding as expected, if I think of the position to be related to the structure of the data and that the model learns such structure, especially if the training sequences are of the same length? In other words, some failure modes might be a natural consequence of the training recipe.
- I'd suggest more clearly separating representational considerations and optimization challenges.
    - Sec 8.4: about increasing the width/depth/heads: do you think the gain results more from optimization or representational benefits?
    - Please consider adding a discussion on the relation to the literature on length generalization and even OOD challenges in general.

**Clarifications**:
- Sec 4.2.3: I don't understand what "natural impulse" means.
- Sec 6.1.2: to clarify, do you use all $2^n$ binary strings during training for sequences of length $n$? Or is $2^n$ merely an upper bound?
- Sec 6.2: there's a typo in the example in the first paragraph?
- Sec 7.1: what are the criterions to determine whether an attention head is a recursive head?
- Sec 7.4: what does it mean by "the weaker notion of executing the algorithm"?
- Sec 8.4, "transformers have demonstrated the capability to learn to parse reductions of a specific number of steps": could you provide references please?
- Sec 10: what does "the model summarized program" mean?
- Sec 11: I'm not sure Perez et al. 2021 is closely related to the narrative of this paragraph, as Perez et al. uses repeated applications of a constant-depth Transformer, whereas this paragraph talks about a varying-depth Transformer (whose depth grows with the recursive depth).

**Writing**: I appreciate it that the paper provides sufficient background and is self-contained. However, some discussions (e.g. some examples in Sec 3 & 4, and details in Sec 6) could be moved to the appendix to make the main paper cleaner to read. The empirical findings could be summarized more crisply.

---

> ### Author Response · Authors · 2024-04-30
> **Response Part 1 of 2**
>
> We thank the reviewer for their feedback, and we are very excited to engage in the discussion with the reviewer on the thought-provoking questions. We address the concerns and provide our thoughts below. Also, we will reflect the reviewer’s suggestions on the next revision.
> # ASM, syntax, and semantics
> ## ASM as a part of semantics or syntax?
> A: An ASM description of a program belongs to the domain of operational semantics, where one analyzes (or reverse-engineers) a program by constructing a model of a computer (or a virtual machine) on which the program runs and describing how the program “runs” on this virtual machine. The structure of the virtual machine model reflects the chosen level of abstraction. This would involve interpreting the syntax of the program as a data-dependent sequence of instructions, which can be thought of as state transformations of an ASM. Replacing natural numbers with dinosaurs may be thought of as replacing semantics while preserving syntax only if this replacement would also affect the abstract interpreter that “runs” the program and thus generates its operational semantics. As long as the functional form of the instruction updates is not altered, the replacement of natural numbers with dinosaurs would leave the operational semantics invariant.
> Mechanistic Interpretability and ASM
> ## Q: ASM and Mechanistic Interpretability
> A: Formally drawing a connection between mechanistic interpretability and the ASM framework is not the main point of discussion in this paper. Instead, our comment on the relation between the two was meant to connect the understanding of operational semantics of a program implemented by a learned neural network with mechanistic interpretability analyses. The relation between mechanistic interpretability and program reverse-engineering by viewing the Transformer architecture as a virtual machine was emphasized by Chris Olah in “Mechanistic Interpretability, Variables, and the Importance of Interpretable Bases” (see https://www.transformer-circuits.pub/2022/mech-interp-essay). This is in accord with the distinction between the denotational semantics and the operational semantics of a program, where the former describes the function implemented by the program from the input-output (external) point of view, while the latter provides a “reverse-engineered” internal view through the lens of a particular type of a virtual machine. In the context of neural nets, the trained Transformer provides the denotational semantics, and the goal of mechanistic interpretability analysis is to furnish an operational semantics grounded in the Transformer architecture. Moreover, the ASM framework is rather flexible and may allow for rather complicated primitives; one of the examples of an ASM given in the literature (W. Reisig, “On Gurevich’s theorem on sequential algorithms,” Acta Informatica 39, 273–305 (2003)) is ruler-and-compass constructions in high school geometry. Here, the set of operations is rather broad, but a typical geometric construction is a sequence of operations obeying the ASM axioms. We shall revise the phrasing of our comment to make these relations more precise.
> # Discussion on Decomposition
> We are excited to share our thoughts on this matter, and we believe this is an interesting problem to investigate. In the context of a pre-trained language model, one potential middle ground could be to teach the model with reinforcement learning, for example. By providing the signal on whether the result of a chain of stepwise reduction is correct, we could teach the policy model (i.e. the model we are training to perform the task) how to perform such reductions without supervising it with the ground-truth reduction chains for every example.
> Some remotely related literature could be [1,2]
>
> Essentially, the intuition behind this idea is to let the model explore and find the optimal strategy for itself to compute the final output, utilizing the knowledge and basic capabilities it gained during pre-training.  And this is not limited to the problem of computing the output of a recursive function, but could potentially be applicable to various problems that require multi-step computation.
>
> [1] Orca-Math: Unlocking the potential of SLMs in Grade School Math ( https://arxiv.org/abs/2402.14830)
>
> [2] Advancing LLM Reasoning Generalists with Preference Trees (https://arxiv.org/abs/2404.02078).

---

> ### Author Response · Authors · 2024-04-30
> **Response Part 2 of 2**
>
> # Positional Encoding
> We thank the reviewer for discussing this with us.
>
> As a clarification, we trained the model on sequences of various lengths and randomly offsetted positional encodings to eliminate the bias brought by positional encodings.
>
> That being said, we agree with the reviewer that it is unsurprising for the model to learn an algorithm that relies on positional information for the highly structured task of computing binary successor. The root cause of the failures on edge cases is the incorrect algorithm the model learns based on that positional information, rather than the fact that the model utilizes such information alone. We will make this point clear to avoid confusion.
>
> # Representational versus Optimization Concerns
>
> We thank the reviewer for initiating this discussion. Our investigation presented in the paper did not aim to distinguish between optimization and representation. Nevertheless, we are excited to share our thoughts based on the existing evidence.
> Attributing the performance increase in tree traversal as model capacity increases:
> As discussed in the paper, the model learns fixed-depth solutions to traversals. As the maximum depth of trees increases, models need to handle more different depths and thus need more heads to be specialized to each. Indeed, as we increase the number of heads given the same embedding size, the performance increases. From this line of reasoning, we conjecture that representation is more likely to be the answer.
> More discussion on generalization-related literature: we thank the reviewer for the suggestion and will include more discussions during revision.
>
> # Clarifications
> We appreciate the reviewer’s effort in helping identify the unclarities in our writing and will make sure these points are presented with better clarity in the revision.
>
> **Number of training sequences for binary successor task**: We did use all 2**n pairs in our main experiment.
> Typo: Yes, it should be (X1 X0 X1 X1 01)= X0 ( X0 X1 X1 01) = X0 X1 X1 X1 01.
>
> **Determining “recursion-head”**: we thank the reviewer for clarification. We use the term “recursion-head” to refer to those decoder self-attention heads whose attention values were concentrated on the bit before recursion starts for natural order and
> Last bit of recursion, for reversed order.
>
> **The weaker notion of executing the algorithm**: Apologies for the phrasing issue. Based on our perturbation analysis, model trained with smaller LRs still exhibit the same algorithmic behavior (i.e. the algorithm they learned is still the same as large LR cases), however, the precision of their execution is lower reflected by a lower success rate and weaker generalization.
>
> **Reference to Perez et al. 2021**: Here is the main point of argument for this section - since transformers are shown to be turing complete by Perez et al., it follows that there exists a configuration that can achieve all computational tasks. Assuming bounded length D for input, we claim that most D layers are needed for the binary successor task.
>
> **Model-summarized program**: We prompted the language models to first summarize the algorithm it mined from examples. This refers to the programs the LLMs ‘think’ to be the underlying algorithm that maps the input to output.

---

### Review · Reviewer_iF1k · 2024-04-22

**Summary Of Contributions:**

This paper presents a comprehensive framework for investigating the ability of transformer-based models to learn structural recursion from examples. The core contributions are twofold:

* First, the authors introduce a general framework for representing and reasoning about structural recursion with sequence models. The framework supports syntax through a sequential encoding based on inductive representations of recursively defined datatypes. It supports semantics through both a stepwise reduction semantics that connects the syntax to traditional programming language semantics, and an Abstract State Machine (ASM) semantics that enables interpreting and analyzing the algorithmic behavior of learned models in implementing those semantics, even when the models are not architecturally recursive. The framework takes into account the complexities of analyzing transformer models, which are not architecturally recursive, and the added intricacies introduced by the knowledge and capabilities acquired during pre-training in large language models.

* Second, guided by their framework, the authors conduct an empirical investigation of sequence model behavior on structurally recursive tasks. They instantiate the framework to variants of two concrete tasks: learning the binary successor function and learning various tree traversals. For each task, they design and conduct experiments to understand the behavior of small transformer models trained from scratch. They reconstruct the learned algorithms and identify specific non-recursive shortcuts the models take, reverse engineer failure cases, and provide mitigations. Additionally, they briefly investigate the ability of pre-trained language models to learn recursive algorithms through fine-tuning and in-context learning. The results suggest transformer models trained on input-output pairs do not fully capture recursion semantics and instead fit shortcuts pruned to edge cases. The models also struggle to mine and execute recursive algorithms from a few in-context demonstrations.

Overall, the paper takes important steps towards understanding how to better handle recursion with sequence models, helping bridge the theory and practice of learning recursion. The framework and empirical results pave a path towards more reliable performance on the many tasks for which recursion is fundamental.

**Audience:**

Yes

**Claims And Evidence:**

Yes

**Requested Changes:**

Please check the weakness part

**Strengths And Weaknesses:**

**Strength**

The paper has several notable strengths:

* The proposed framework is well-motivated, comprehensive and general. It provides a principled way to investigate structural recursion in sequence models by cleanly connecting abstract recursive concepts to concrete sequence modeling problems and model behaviors. The framework supports both syntax and semantics, and the dual semantic models (reduction and ASM) enable reconciling symbolic and neural behavior.
* The framework is thoughtfully and flexibly designed to handle complexities like the non-recursive nature of transformers, the impacts of pre-training in large language models, and in-context learning.
* The two chosen tasks - binary successor and tree traversal - are simple yet interesting, and capture different types of recursive structure. They serve as strong test cases for the framework.
* The empirical methodology is thorough and multi-faceted. For each task and model, the authors examine representation capacity, execution ability, attention patterns, failure modes, and more. They employ techniques like perturbation analysis, counterfactual patching, and probing to reverse engineer learned algorithms.
* The results are insightful and impactful. Key findings include: models learn non-recursive shortcuts that fail on edge cases; it's difficult for models to learn recursive patterns from a few examples; and models can perform partial recursion but struggle to fully capture semantics. Practical mitigations like data balancing are discussed.
* Connections to theory and practice are emphasized throughout. The authors relate empirical results back to their theoretical framework, discuss gaps between theory and practice, and highlight how the work can guide more reliable use of sequence models for recursive tasks.
The paper is comprehensive, with many additional results and analyses provided in the appendix. Yet the main body is kept streamlined and digestible.

In summary, this is an impressive paper that advances our understanding of learning recursion and provides a solid foundation for future work at the intersection of sequence modeling and structural recursion. The framework is well-designed and general, the empirical investigations are extensive and insightful, and the discussion connects theory and practice.

**Weakness**

While the paper is strong overall, there are a few areas that could be improved:

* The framework is complex and takes significant effort to fully understand. The connections between the syntactic encoding, reduction semantics, ASM semantics, and recursive/non-recursive model implementations are conceptually dense. More explanatory prose and figures could help make the framework more accessible to a broader audience. That said, given the inherent complexity of the topic, the authors do a commendable job laying out the framework piece by piece.
* The empirical results, while extensive, can feel a bit scattered at times. The sheer number of model variations, task set-ups, and analysis techniques used makes it difficult to maintain a cohesive narrative throughout the experiments section. The authors could consider restructuring to prioritize the most important results and provide clearer takeaways before diving into additional variations. However, the overall empirical methodology is sound and justified given the goal of comprehensively stress-testing model capabilities.
* Some of the experiments, such as in-context learning with very large language models, are less conclusive than others due to computational and API constraints. It would be ideal to further scale up these experiments, but the preliminary results still provide valuable insights and lay groundwork for future work. The authors are transparent about experimental limitations.
* The paper is very long and dense, which may hinder broad readership. However, the length is justified by the scope of the contributions - the framework is nuanced and the empirical investigations are extensive. The authors balance this by keeping the main body focused and deferring many details to the appendix.
* Societal impact and future directions could be discussed in more depth. However, this is understandable given the conceptual nature of the contributions, and the authors do touch on these points in the conclusion. Future work could further explore how the framework can translate to more reliable deployment of recursive models in real-world applications.

In summary, the weaknesses are relatively minor and do not significantly detract from the paper's contributions. The main points that could be improved are: 1) Making the dense theoretical framework more accessible, 2) Streamlining the presentation of empirical results, 3) Strengthening experiments that were limited by computation. However, the authors have clearly put a lot of thought into balancing the paper's complexity with explanatory clarity, and the results are significant despite any experimental constraints. Overall this is strong work that will likely have substantive impact on the field.

---

> ### Author Response · Authors · 2024-04-30
>
> First of all, we thank the reviewer for the constructive and helpful comments! We are glad and highly encouraged that the reviewer finds the work interesting and meaningful! We respond to the reviewer’s comments below.
> # Framework
> We appreciate your understanding of the complexity of the topic and of our attempts to explain the framework despite that. We will do our best to continue making the framework more accessible to a broader audience using your suggestions.
>
> # Improving the presentation of empirical findings
> We will do our best to organize the empirical results better using your suggestions, for example by providing clearer takeaways early on.
> We will also consider any suggestions from all the reviewers to improve the presentation for a broad audience.
> # Future work
> We thank the reviewer for understanding the computational constraints and also appreciate their interest in seeing more interesting results and insights. We will continue this line of investigation and incorporate more insightful results.
>
> One direction we could further go into is to understand better how models emulate program execution in general. Another possible direction is to continue investigating such recursive computation and see whether the transformers lack such capability and whether this further reveals its limitations.
>
> We will add a future work section and discuss the possibilities of future research questions. Also, we will discuss societal impacts in the paper.

---

### Decision · Action_Editor_8fEu · 2024-06-17

**Recommendation:** Accept with minor revision

**Comment:**

Reviewers agreed that the paper tackles an important research question. Two reviewers were most positive and further highlighted how the paper is well designed, bridges theory and practice and provides fruitful insights on how to investigate what complex ML models can learn through formal PL methods. At the same time, they raised a number of concerns pertaining the dense writing (the material is squeezed into ~34 pages + 10 pages appendix, but could likely benefit from more), the scattering of experimental results and the lack of a clear-cut message after some experiments (adding recap boxes and takeaway messages can help).

One reviewer has been skeptical about the claims of the paper and voted towards rejecting it considering that the two example scenarios analyzed are simple and might be too far from real-world reasoning scenarios. I agree on the simplicity of the scenarios investigated, but disagree that this invalidates the claims (one needs only one counterexample to disprove a "common hypothesis" which in this case is that Transformers can flawlessly handle recursion).

Authors in the rebuttal phase addressed the most important aspects in the comments, clarifying the major concerns of two reviewers. However, they did something "unorthodox" for TMLR by not uploading a revised version of the manuscript that includes all the promised changes. I am recommending acceptance nevertheless, as two reviewers were still happy after the discussion. Please note however that this *acceptance is conditional* on the fact that the authors implement all the reviewers' requests (especially APjT and iF1k) in their minor revision. These requests are not optional and the paper will only be published if all the comments have been addressed in the new revision. Also note that TMLR does not have multiple rounds of revision as other journals.

**Audience:**

The topic is relevant for the TMLR audience as revolves around understanding what some of the most successful architectures in deep learning can (and cannot) learn well from data.

**Claims And Evidence:**

The paper deals with the important question of understanding how Transformers can handle recursion. The authors propose a general framework to represent and reason about recursion over sequences, and define its proper syntax and semantics in a classical programming languages (PL) fashion. Under this framework it is possible to analyze models such as Transformers that do not seem to be designed to support structural recursion.

Furthermore, this framework helps highlight what to investigate empirically to find practical shortcomings when learning with recursion over sequences. Specifically, the authors instantiate the framework to two simple tasks: learning to traverse trees and learning the binary successor function and they highlight how the models can take shortcuts without recursion.

While these are simple learning scenarios, they are enough to substantiate the claim that Transformer-based models do not fully capture recursion and paves the way to find solutions to this problem.